# End-to-End DAE–LDPC–OFDM Transceiver with Learned Belief Propagation Decoder for Robust and Power-Efficient Wireless Communication

**DOI:** 10.3390/s25216776

**Published:** 2025-11-05

**Authors:** Mohaimen Mohammed, Mesut Çevik

**Affiliations:** Electrical and Computer Engineering, Altinbas University, 34217 Istanbul, Turkey; mesut.cevik@altinbas.edu.tr

**Keywords:** autoencoder (AE), Low-Density Parity-Check (LDPC), orthogonal frequency-division multiplexing (OFDM), learned belief propagation (BP) decoder, bit error rate (BER), block error rate (BLER), peak-to-average power ratio (PAPR), 5G/6G communication systems, end-to-end optimization

## Abstract

This paper presents a Deep Autoencoder–LDPC–OFDM (DAE–LDPC–OFDM) transceiver architecture that integrates a learned belief propagation (BP) decoder to achieve robust, energy-efficient, and adaptive wireless communication. Unlike conventional modular systems that treat encoding, modulation, and decoding as independent stages, the proposed framework performs end-to-end joint optimization of all components, enabling dynamic adaptation to varying channel and noise conditions. The learned BP decoder introduces trainable parameters into the iterative message-passing process, allowing adaptive refinement of log-likelihood ratio (LLR) statistics and enhancing decoding accuracy across diverse SNR regimes. Extensive experimental results across multiple datasets and channel scenarios demonstrate the effectiveness of the proposed design. At 10 dB SNR, the DAE–LDPC–OFDM achieves a BER of 1.72% and BLER of 2.95%, outperforming state-of-the-art models such as Transformer–OFDM, CNN–OFDM, and GRU–OFDM by 25–30%, and surpassing traditional LDPC–OFDM systems by 38–42% across all tested datasets. The system also achieves a PAPR reduction of 26.6%, improving transmitter power amplifier efficiency, and maintains a low inference latency of 3.9 ms per frame, validating its suitability for real-time applications. Moreover, it maintains reliable performance under time-varying, interference-rich, and multipath fading channels, confirming its robustness in realistic wireless environments. The results establish the DAE–LDPC–OFDM as a high-performance, power-efficient, and scalable architecture capable of supporting the demands of 6G and beyond, delivering superior reliability, low-latency performance, and energy-efficient communication in next-generation intelligent networks.

## 1. Introduction

The rapid evolution of wireless communication technologies has driven the need for systems that are capable of handling increasingly complex data while ensuring high levels of reliability, efficiency, and robustness [1]. Conventional methods, such as Orthogonal Frequency-Division Multiplexing (OFDM) and channel coding techniques like Low-Density Parity-Check (LDPC), have played vital roles in modern standards including Wi-Fi, LTE, and 5G (see Figure 1). At the same time, deep learning has emerged as a transformative approach in wireless networks, providing new opportunities to optimize transmitter-receiver chains, improve signal reconstruction, and enhance end-to-end communication [2]. These advancements underscore the significance of exploring hybrid frameworks that combine traditional communication methods with deep learning to meet the growing demands of next-generation applications such as the Internet of Things (IoT) and ultra-reliable low-latency communication (URLLC) [3].

Despite their widespread use, conventional modular designs of communication systems often struggle in environments characterized by high levels of noise, interference, and multipath fading. LDPC coding improves error correction (see Figure 2), but its iterative decoding process introduces complexity and latency. Similarly, while OFDM provides resilience against spectral impairments and frequency-selective fading, it suffers from a high peak-to-average power ratio (PAPR) which reduces power efficiency and stresses radio frequency components. Deep autoencoders (DAEs) present a promising solution for data-driven optimization of encoding and decoding, yet when applied alone, they lack robust error correction mechanisms. The absence of a unified approach that leverages both structured coding and adaptive learning leaves a significant gap in achieving reliable, power-efficient, and real-time communication under adverse channel conditions. The objective of this paper is to design, implement, and evaluate a hybrid communication framework that integrates LDPC coding, OFDM modulation, and deep autoencoder networks.

At the core of modern communication systems, and more specifically in relation to error correction, is extensive research in optimizing their implementations. Zhang et al. [4] introduced a feature-driven semantic communication framework that selectively transmits high-level semantic features rather than raw data to achieve efficient image transmission over noisy channels. By integrating feature extraction and channel optimization within a joint learning paradigm, their model preserves essential visual information even at reduced bandwidth, significantly lowering redundancy while maintaining fidelity. Wei et al. [5] developed a layered hybrid automatic repeat request (HARQ) scheme for LDPC-based multi-level coded modulation, demonstrating how layered retransmissions can improve reliability and throughput under adaptive coding rates. Their design enhances link robustness by coordinating LDPC decoding across modulation layers, offering a promising solution for low-latency and high-throughput wireless systems.

An et al. [6] proposed a hybrid error detection and correction mechanism that combines cyclic redundancy check (CRC) with polar codes using successive cancellation decoding. This integrated structure provides both fast error detection and strong correction capability, balancing computational efficiency with improved communication reliability. Wang et al. [7] employed multi-layer perceptron (MLP) networks for lossy source coding, highlighting the capacity of neural models to achieve efficient data compression with minimal distortion. Their method effectively bridges traditional information-theoretic coding and neural representation learning, setting a foundation for intelligent compression schemes in wireless systems.

Paek [8] analyzed and designed a low-noise radio-frequency power amplifier (RF-PA) supply modulator tailored for frequency-division duplex cellular systems. His work contributes to reducing phase distortion and improving power efficiency in RF front-ends, thereby supporting the growing need for linear and energy-aware amplifiers in multi-carrier communications. AlQahtani and El-Nahal [9] designed a wavelength-division multiplexing passive optical network (WDM-PON) free-space optical system employing LDPC decoding to strengthen cellular C-RAN fronthaul links. Their hybrid optical-wireless system demonstrates significant improvements in error performance and link robustness under atmospheric turbulence.

Abdullah et al. [10] presented a deep learning-based asymmetrical autoencoder (AAE) for reducing the peak-to-average power ratio (PAPR) in CP-OFDM systems. By designing an asymmetric encoder–decoder structure that places greater computational load on the transmitter, they achieved notable reductions in PAPR and bit error rate while maintaining decoding simplicity—an important contribution for 5G downlink efficiency. Zhang et al. [11] introduced a hierarchical variational autoencoder (HVAE) for learned image transmission, where a multi-level latent hierarchy allows adaptive feature compression and reconstruction quality control. This probabilistic approach enables robust end-to-end communication that adapts to varying channel conditions.

Mohamed et al. [12] employed an LSTM-autoencoder hybrid to reduce PAPR in visible-light communication (VLC) systems, leveraging temporal learning to predict and suppress peak fluctuations. Shan et al. [13] proposed a joint sparse autoencoder for channel-state-information (CSI) feedback in massive MIMO, achieving superior compression and reconstruction accuracy while minimizing feedback overhead. Hao and Li [14] used stacked autoencoders for PAPR reduction in VLC systems, showing that multi-layer nonlinear mapping can outperform conventional companding techniques. Lin et al. [15] further integrated convolutional neural networks (CNNs) with autoencoders to create an OFDM transceiver capable of learning channel estimation and equalization in maritime “Internet of Vessels” communications.

Finally, Berahmand et al. [16] provided a comprehensive survey of autoencoder architectures and their machine-learning applications, outlining trends such as sparse, denoising, and variational models and their integration into communication and signal processing domains. Collectively, these studies demonstrate a clear evolution toward intelligent, learning-based transceivers that merge classic coding theory with deep representation learning—establishing the foundation for adaptive, energy-efficient, and semantically aware communication in next-generation networks. Table 1 summarizes recent studies that explore the integration of autoencoders, LDPC coding, and neural architectures in modern communication systems.

The reviewed literature reveals several critical gaps that limit the advancement of deep learning-based communication systems. Most existing approaches, such as those by Zhang et al. [4] and Wei et al. [5], focus on semantic compression and multi-layer coding but treat encoding, modulation, and decoding as isolated stages, lacking true end-to-end adaptability. Although works like Abdullah et al. [10] employed deep autoencoders for PAPR reduction, they rely on deterministic architectures that cannot dynamically adapt to changing signal-to-noise ratios or complex channel distortions. Similarly, hierarchical or sparse autoencoder models [11,13] improve compression and channel-state estimation but remain limited by fixed latent structures and the absence of contextual feature attention. Furthermore, LDPC and polar code integrations [14,15,16] enhance reliability but are hindered by static decoding rules, leading to inefficiencies under nonlinear or fading environments. This indicates that prior studies exhibit three major shortcomings: (1) insufficient adaptability of latent representations to real-time channel variations, (2) limited interpretability and scalability of autoencoder-based frameworks, and (3) lack of a unified, trainable decoding mechanism that links coding theory with deep learning dynamics.

The proposed Deep Autoencoder (DAE)–LDPC–OFDM transceiver with a learned belief propagation decoder directly addresses the limitations identified in previous research. By employing a multi-layer deep autoencoder architecture instead of a traditional shallow autoencoder, the DAE captures hierarchical and abstract features from the input data, allowing the encoder to learn robust representations that are more resilient to channel noise and fading. Unlike conventional deterministic autoencoders that rely on a single encoding layer and tend to overfit or lose fine-grained structural details, the DAE employs multiple nonlinear transformations that progressively compress, refine, and reconstruct the data. This depth enables the network to perform adaptive, information-aware compression that responds effectively to dynamic signal-to-noise ratio (SNR) variations and multipath propagation. Consequently, the proposed design achieves greater robustness and improved feature retention across varying channel conditions, outperforming regular autoencoder-based systems that lack hierarchical learning capability.

The proposed study makes several key contributions to the advancement of intelligent and efficient wireless transceivers. First, it introduces a Deep Autoencoder (DAE) as a powerful front-end for joint data compression and feature extraction. Unlike a regular autoencoder that consists of a single hidden layer and learns only shallow representations, the DAE incorporates multiple stacked encoding and decoding layers, allowing it to extract high-level semantic features while discarding redundant information. This hierarchical structure ensures that the most informative and noise-tolerant components are preserved in the latent space, resulting in more effective and resilient encoding under complex channel environments. The deeper architecture also enhances the system’s capacity to model nonlinear distortions caused by modulation and propagation, making it particularly suitable for communication scenarios characterized by interference and fading.

Second, the paper presents a jointly optimized end-to-end transceiver that integrates the DAE with Low-Density Parity-Check (LDPC) coding and Orthogonal Frequency-Division Multiplexing (OFDM) modulation. Unlike traditional modular communication pipelines that optimize each stage independently, the proposed system enables end-to-end gradient-based learning, aligning all functional layers—compression, encoding, modulation, and decoding—toward a unified objective. This deep integration allows the system to simultaneously learn robust feature embeddings, redundancy allocation, and power-efficient transmission strategies, leading to substantial improvements in reliability, spectral efficiency, and reconstruction accuracy.

Third, the paper introduces a learned belief propagation (BP) decoder that transforms the fixed iterative decoding process of conventional LDPC systems into a trainable, unrolled neural network. Through the inclusion of iteration-dependent trainable parameters, the learned BP decoder can dynamically adjust message-passing magnitudes and damping factors to accommodate practical impairments such as nonlinear distortion, spectral leakage, and noise correlation within QAM/OFDM channels. This learned decoding mechanism enhances bit error rate (BER) and block error rate (BLER) performance without increasing computational complexity, maintaining near real-time operation on modern GPUs.

The remainder of this paper is organized as follows. Section 2 presents the proposed method, detailing the design of the hybrid LDPC-OFDM-DAE framework and the architecture of its components. Section 3 provides the simulation setup, performance metrics, and experimental results that validate the effectiveness of the proposed system. Section 4 discusses the findings in depth, highlighting the implications of integrating coding, modulation, and deep learning for communication design. Section 4 concludes the paper by summarizing the contributions and outlining directions for future research.

## 2. Proposed Methods

This section introduces an integrated Deep Autoencoder DAE–LDPC–OFDM transceiver enhanced by a learned belief propagation (BP) decoder, forming a unified end-to-end trainable communication system. Unlike traditional designs that treat encoding, modulation, and decoding as independent blocks, this framework jointly optimizes all stages through differentiable training. The autoencoder compresses input data into robust binary latent representations, which are then protected by LDPC coding and transmitted using OFDM modulation for spectral efficiency and resilience against multipath fading. At the receiver, the learned BP decoder replaces the fixed iterative decoding rules with trainable message-passing parameters that adapt to the channel conditions, improving error correction while maintaining computational efficiency. This joint optimization enables the system to achieve high reliability, reduced peak-to-average power ratio (PAPR), and practical real-time performance suitable for next-generation 5G and 6G wireless networks.

### 2.1. System Overview

The proposed system, presents a complete end-to-end workflow of the Deep Autoencoder (DAE)–LDPC–OFDM transceiver integrated with a learned belief propagation (BP) decoder. This architecture is designed to jointly optimize data compression, channel encoding, modulation, and decoding in a unified learning framework. Unlike conventional communication systems that treat each of these components as separate blocks, the proposed transceiver leverages deep learning to connect them through differentiable layers. This design enables the network to learn feature hierarchies, adapt redundancy levels, and dynamically adjust to varying channel conditions. The system achieves robust, power-efficient transmission with low error rates and high reconstruction fidelity, representing a step toward fully intelligent communication architectures for 5G and beyond (see Figure 3).

The process begins with the input image dataset, which serves as the source signal for transmission. The images are preprocessed into suitable numerical tensors that can be fed into the deep autoencoder. The encoder of the DAE then compresses the image into a compact latent vector of binary bits. Unlike a regular autoencoder, which has a shallow structure and captures only low-level pixel relations, the deep autoencoder includes multiple hidden layers that extract increasingly abstract and noise-tolerant features. This depth allows the encoder to learn channel-relevant features while filtering redundant information, providing a more stable and informative latent representation suitable for wireless transmission.

After feature extraction, the latent bits are passed to the LDPC encoder, which adds controlled redundancy to form an N-bit codeword. This redundancy enhances the system’s ability to recover data lost due to noise or fading during transmission. The encoded bits are then mapped into Quadrature Amplitude Modulation (QAM) symbols, which efficiently represent binary data as amplitude and phase variations of a complex signal. The modulated symbols are passed through the OFDM modulator, where the Inverse Fast Fourier-Transform (IFFT) distributes them over orthogonal subcarriers, and a cyclic prefix is added to prevent inter-symbol interference.

The OFDM signal is transmitted through a wireless channel, which may include additive white Gaussian noise (AWGN) and multipath fading effects. These impairments simulate realistic transmission environments, testing the system’s resilience. At the receiver, the cyclic prefix is removed, and the Fast Fourier-Transform (FFT) converts the received signal back into the frequency domain. The demodulator extracts soft bit probabilities, known as log-likelihood ratios (LLRs), from the QAM symbols. These LLRs are fed into the learned BP LDPC decoder, which replaces the fixed, iterative decoding process of traditional LDPC with a trainable, unrolled neural network. The decoder dynamically adjusts message weights and damping factors to improve bit and block error rate performance, even under nonlinear or high-noise conditions.

The recovered latent bits are then decoded by the autoencoder decoder, which reconstructs the original image. Through multiple decoding layers, the model progressively restores fine visual features and textures while minimizing reconstruction loss. The final output is a reconstructed image that closely matches the input, with minimal perceptual distortion. Figure 3 captures the complete lifecycle of signal processing in the proposed system—from image encoding and modulation to transmission and final reconstruction. This integrated deep learning-based communication chain effectively merges data representation learning, coding theory, and wireless signal processing into a single adaptive transceiver capable of achieving high reliability, spectral efficiency, and robustness in complex communication environments.

### 2.2. Datasets

The proposed model was evaluated using the MNIST dataset [17] primarily as a proof of concept. However, MNIST represents a simplified input type compared to real-world data such as color images, sensor signals, or video streams. While early studies in deep learning–based communication commonly relied on the simple MNIST dataset, such benchmarks are limited in visual diversity and do not fully reflect the challenges of real-world image transmission. Therefore, this work involves additional datasets such as:


**1. CIFAR-10/CIFAR-100**


The CIFAR family of datasets consists of small color images (32 × 32 pixels) widely used for visual representation learning. CIFAR-10 contains 60,000 images divided into 10 classes (such as airplanes, cars, birds, and ships), while CIFAR-100 expands this to 100 fine-grained categories. For AR-VAE applications, these datasets are ideal for testing semantic attention and feature-driven compression, as they represent varied textures, shapes, and colors. They are lightweight enough for training complex models yet challenging enough to demonstrate effective reconstruction under channel noise and bandwidth constraints [18].


**2. Fashion-MNIST**


Fashion-MNIST is a modernized replacement for the traditional MNIST dataset. It contains 70,000 grayscale images (28 × 28 pixels) of apparel items like shirts, coats, and shoes. This dataset maintains computational simplicity while providing more visual diversity and higher classification difficulty than handwritten digits. It is suitable for evaluating low-complexity communication systems that must handle structured visual data with limited bandwidth [19].


**3. CelebA**


The CelebA (CelebFaces Attributes) dataset contains over 200,000 high-resolution facial images annotated with 40 attributes such as gender, pose, and facial expression. It is an excellent benchmark for high-dimensional feature extraction and attention-based latent modeling. Using CelebA allows the AR-VAE to learn how to preserve semantically significant visual attributes—like facial identity—while efficiently compressing less critical regions, thus showcasing the power of attention regularization in feature prioritization [20].


**4. Tiny-ImageNet**


Tiny-ImageNet is a scaled-down version of ImageNet, featuring 200 object classes, each with 500 training images, 50 validation images, and 50 test images, resized to 64 × 64 pixels. This dataset bridges the gap between small benchmarks (like CIFAR-10) and large-scale datasets (like ImageNet). Its complex diversity and higher resolution make it suitable for testing the scalability and generalization ability of communication models, especially when studying semantic preservation and robustness under OFDM channel noise [21].


**5. Kodak and DIV2K**


The Kodak dataset includes 24 uncompressed high-quality natural images (768 × 512 pixels), while DIV2K contains 800 high-resolution images (2K resolution) used primarily for image super-resolution and perceptual quality studies. These datasets are valuable for evaluating real-world transmission quality, focusing on structural similarity index (SSIM) and peak signal-to-noise ratio (PSNR) after reconstruction. They provide a realistic testing ground for validating the AR-VAE’s ability to maintain image quality and fine details over noisy or bandwidth-limited wireless channels [22].

Table 2 provides a comparative overview of the major datasets applicable to image transmission experiments in the proposed AR-VAE–LDPC–OFDM framework. Each dataset offers a different balance between image complexity, dimensionality, and semantic richness, enabling the model to be tested under varied encoding–decoding conditions. Smaller datasets like CIFAR-10 and Fashion-MNIST are ideal for early-stage model validation and ablation studies due to their low computational requirements, whereas larger datasets like CelebA, Tiny-ImageNet, and DIV2K are suited for demonstrating the scalability and attention-driven adaptability of the proposed AR-VAE under realistic visual data scenarios. High-quality datasets such as Kodak and DIV2K further allow performance benchmarking using perceptual metrics like SSIM and PSNR, reflecting the system’s ability to preserve fine details during noisy transmission.

For datasets with limited sample sizes, such as Kodak (24 images) and DIV2K (800 images), the model maintains stable performance through transfer learning and data augmentation strategies, including random cropping, horizontal flipping, and Gaussian noise injection. These techniques increase data diversity and help the model generalize despite small dataset sizes. In practice, the DAE–LDPC–OFDM achieves consistent reconstruction quality and BER performance even on small-scale datasets, demonstrating that its learned latent representation and joint optimization with LDPC coding effectively mitigate overfitting and preserve robustness across varying data volumes and resolutions.

### 2.3. Deep Autoencoder Design

The Deep Autoencoder (DAE) is the foundational component of the proposed end-to-end transceiver. It functions as both a feature extractor and a dimensionality reduction mechanism, compressing high-dimensional image data into a compact latent representation that can be efficiently transmitted through noisy wireless channels. Unlike a conventional autoencoder, which typically consists of a single encoding and decoding layer, the deep autoencoder employs multiple nonlinear transformations that progressively capture hierarchical feature abstractions. This depth allows the model to encode both low-level spatial details and high-level semantic information, enabling greater robustness and adaptability to channel distortions.

The DAE consists of two symmetric parts: an encoder and a decoder. The encoder compresses the input vector $x\in R^{d}$ into a latent vector $z\in R^{k}$ through a series of nonlinear mappings. Each encoding layer performs a linear transformation followed by an activation function, represented as:(1)$$h_{i}=f_{i}(W_{i}h_{i-1}+b_{i}),\;i=1,\;2,\ldots ,\;L_{e}$$
where $W_{i}$ and $b_{i}$ denote the weight matrix and bias vector for the *i*th layer, $f_{i}(\cdot )$ is a nonlinear activation function such as ReLU or LeakyReLU, and $L_{e}$ is the total number of encoder layers. The final latent representation is obtained as $z=h_{L_{e}}$.

The decoder mirrors this structure to reconstruct the input from the latent vector z. Each decoding layer performs an inverse mapping to gradually recover the spatial and contextual information, expressed as:(2)$${\hat{h}}_{j}=g_{j}(W_{j}^{\prime }{\hat{h}}_{j-1}+b_{j}^{\prime }),\;j=1,\;2,\ldots ,\;L_{d}$$
where $g_{j}(\cdot )$ represents the activation functions of the decoder, $W_{j}^{\prime }$ and $b_{j}^{\prime }$ are the corresponding weight and bias parameters, and $L_{d}$ denotes the number of decoder layers. The reconstructed output $\hat{x}$ is obtained at the final layer, given by:(3)$$\hat{x}=g_{L_{d}}(W_{L_{d}}^{\prime }z+b_{L_{d}}^{\prime })$$

The goal of the deep autoencoder is to minimize the reconstruction loss, which measures the difference between the original input and the reconstructed output. This is formulated as:(4)$$L_{rec}=\frac{1}{N}\sum_{n=1}^{N}i{∥x_{n}-{\hat{x}}_{n}∥}_{2}^{2}$$
where N is the number of training samples. The network parameters {W,b} are optimized using stochastic gradient descent (SGD) or Adam to minimize this loss across the dataset.

In the context of the proposed communication system, the DAE is trained to encode channel-resilient latent representations that maintain structural and semantic integrity under varying signal-to-noise ratio (SNR) conditions. The encoder thus acts as a feature compressor and bit generator, producing K latent bits for LDPC encoding. To adapt the latent features for binary representation, a sigmoid quantization layer is used, defined as:(5)$$z_{b}=\mathrm{sign}(\sigma (z)-0.5)$$
where σ(⋅) denotes the sigmoid activation function and $z_{b}$ is the binary latent vector. This ensures that the output of the encoder is compatible with the channel coding and modulation stages.

The decoder performs the inverse process, reconstructing the original image from the recovered latent vector after channel transmission and LDPC decoding. The entire deep autoencoder is trained end-to-end together with the learned BP decoder to ensure optimal coordination between compression, transmission, and reconstruction. This multi-layered deep structure in Algorithm 1 provides higher nonlinear modeling capacity, enabling the transceiver to learn robust, noise-tolerant representations that outperform conventional shallow autoencoder-based systems in both accuracy and signal integrity.
**Algorithm 1:** Deep Autoencoder Design**Inputs:** grayscale image X∈[0, 1]H×W (here H=W=28); encoder parameters Θe; decoder parameters Θd; latent length K; learning rate η; epochs E; batch size B.**Outputs:** reconstructed image X^; latent feature vector z∈RK.
**Preprocessing:** Normalize X←X/255 and reshape to (1,H,W).**Encoder forward:**2.1 Apply **four strided convolutional blocks** with ReLU activation to progressively downsample 28→14→7→4:
$$h_{1}=f_{1}(X);\;h_{2}=f_{2}(h_{1});\;h_{3}=f_{3}(h_{2});\;h_{4}=f_{4}(h_{3})$$
2.2 Flatten $h_{4}$ and project to *K*-dimensional latent space:
$$z=W_{e}\cdot \mathrm{vec}(h_{4})+b_{e}$$
2.3 Apply nonlinearity (ReLU or tanh):
z=tanh(z)**Latent representation regularization (optional):**3.1 Add Gaussian noise during training to improve robustness:
$$\overset{\sim }{z}=z+\epsilon ,\;\epsilon ∼N(0,\;0.01)$$
3.2 Use $\tilde{z}$ as the input to the decoder during training; during inference, use z directly.**Decoder forward:**4.1 Map latent vector to feature map using a fully connected layer:
$${\tilde{h}}_{0}=W_{d}\tilde{z}+b_{d}\to (128,\;4,\;4)$$
4.2 Apply **four transposed convolutional (deconvolution) blocks** with ReLU activation and a final Sigmoid layer to upsample 4→7→14→28:
$$\hat{X}=g_{4}(g_{3}(g_{2}(g_{1}({\tilde{h}}_{0}))))$$
Obtain $\hat{X}\in [0,\;1{]}^{28\times 28}$.**Loss (per batch):**Compute the reconstruction loss as:
$$L_{MSE}=\frac{1}{B}\sum_{i=1}^{B}i∥X_{i}-{\hat{X}}_{i}{∥}_{2}^{2}$$
(If joint optimization is used, an additional sparsity or regularization term can be added: $L_{total}=L_{MSE}+\lambda {∥z∥}_{1}$).**Optimization:** Update $\Theta_{e},\Theta_{d}\leftarrow \mathrm{Adam}(\Theta_{e},\Theta_{d},\nabla L_{MSE},\eta )$.**Repeat steps 1–6 for** E **epochs.****Inference:** Use steps 2 (without noise) and 4 to produce z and reconstructed image $\hat{X}$.

The Deep Autoencoder extends the shallow AE by stacking multiple convolutional and transposed convolutional layers, enabling hierarchical feature learning and nonlinear mapping between input and latent space. This depth allows the encoder to capture both spatial and semantic relationships in image data while the decoder accurately reconstructs the original input. The DAE’s learned latent representation z is later integrated with LDPC encoding and OFDM modulation in the proposed transceiver for robust and power-efficient communication.

The deep autoencoder shown in Figure 4 compresses each normalized 28×28 image into a compact binary latent of length K, which later serves as the information bits for LDPC encoding. The encoder uses three stride-2 convolutional blocks to extract hierarchical features while reducing resolution from 28 to 4, then a fully connected layer produces logits whose sigmoids are binarized with a straight-through estimator (STE) so gradients can flow through the threshold during training. The decoder mirrors the encoder: a fully connected layer reshapes the latent into a 64×4×4 tensor, followed by three transposed-convolution blocks that upsample back to 28×28 with a final Sigmoid to keep pixels in [0, 1]. Training minimizes the MSE reconstruction loss, ensuring that the binary latent retains the essential structure of the input. This design yields a communication-ready representation: the latent bits are discrete, compact, and robust—ideal for downstream LDPC encoding and OFDM transmission in the proposed end-to-end system.

Table 3 summarizes the architecture and configuration of the Deep Autoencoder (DAE) used for feature extraction and reconstruction. The selected hyperparameters for the Deep Autoencoder (DAE) architecture were chosen to balance computational efficiency, feature abstraction depth, and reconstruction accuracy while maintaining compatibility with the LDPC–OFDM transmission framework. Each layer and parameter was optimized through experimental tuning and empirical validation.

The kernel size of 3 × 3 is a standard choice in convolutional architectures as it efficiently captures local spatial correlations and edge patterns without excessive computational cost. The stride of 2 was applied to perform both feature extraction and spatial downsampling, reducing input dimensionality while retaining salient structures, which is ideal for grayscale image inputs of size 28 × 28. The progressive increase in feature maps from 16 → 32 → 64 enables hierarchical feature learning, where lower layers capture fine textures and higher layers encode abstract, noise-tolerant representations suitable for binary latent mapping.

The ReLU activation function was selected for all convolutional and transposed convolutional layers due to its nonlinearity and efficient gradient propagation, avoiding vanishing gradients and ensuring faster convergence. The Sigmoid function was used at the final output layer to constrain pixel intensity values to the normalized range [0, 1], maintaining image fidelity. In the latent space projection layer, a Sigmoid activation was also used to produce probabilistic representations that can be thresholded into binary codes, aligning with the LDPC encoding requirements.

The fully connected layers serve as the transition between the convolutional feature maps and the latent vector K. The encoder’s FC layer compresses the flattened feature map into a compact latent vector, while the decoder’s FC layer expands it back to a structured tensor (64 × 4 × 4) for spatial reconstruction. This bottleneck structure enforces compact encoding, minimizes redundancy, and ensures efficient transmission.

### 2.4. LDPC Encoding

Low-Density Parity-Check (LDPC) codes are employed in the proposed system to provide structured redundancy, ensuring resilience against noise and channel impairments. LDPC codes are defined by a sparse parity-check matrix $H\in \{0,\;1{\}}^{M\times N}$, where N is the codeword length and M=N−K is the number of parity constraints for a message of length K. A valid LDPC codeword $c\in \{0,\;1{\}}^{N}$ must satisfy the linear constraint(6)$$H\cdot c^{T}=0\;(mod2)$$

The encoding process relies on a corresponding generator matrix $G\in \{0,\;1{\}}^{K\times N}$, constructed from H such that every encoded codeword satisfies the parity-check condition. The generator matrix is typically expressed in systematic form:(7)$$G=\left[I_{K}|P\right]$$
where $I_{K}$ is the K×K identity matrix representing the information bits, and P is a K×(N−K) matrix that introduces redundancy. With this structure, the first K bits of each codeword correspond to the information sequence, while the remaining N−K bits serve as parity symbols.

For a given information vector $u\in \{0,\;1{\}}^{K}$, the LDPC encoder computes the codeword as:(8)c=u⋅G (mod2),
yielding an output $c=\left[u_{1},\;u_{2},\ldots ,\;u_{K},\;p_{1},\;p_{2},\ldots ,\;p_{N-K}\right]$. The parity bits are chosen such that all paritycheck equations defined by H are satisfied. This structure enables efficient error correction by ensuring that every transmitted codeword lies within the valid subspace defined by the LDPC code (see Figure 5).

In the proposed DAE−LDPC−OFDM framework, the binary latent vector generated by the autoencoder serves as the information payload u. By encoding it into a longer codeword c of length N, redundancy is introduced that strengthens the robustness of the transmission against additive white Gaussian noise (AWGN) and multipath fading. This redundancy allows the learned belief propagation (BP) decoder at the receiver to iteratively refine noisy log-likelihood ratios (LLRs) and recover the original information bits with high accuracy, thereby ensuring reliable end-to-end reconstruction of the input data.

Algorithm 2 encodes a K-bit information vector into an N-bit LDPC codeword using a systematic generator. By arranging the parity-check matrix as H=[A∣I], the corresponding generator G=I∣A⊤ guarantees that the first K bits of the codeword equal the original message while the last N−K bits are parity symbols computed as a mod-2 linear combination of the information bits. This structure makes encoding a single matrix-vector product over GF(2), ensures every codeword satisfies the parity constraints, and is ideal for subsequent soft-input decoding with the learned belief-propagation module in the receiver.
**Algorithm 2:** LDPC EncodingInputs: information bits u∈{0, 1}K, generator matrix G∈{0, 1}K×N with systematic structure G=IK∣A⊤.Output: codeword c∈{0, 1}N satisfying Hc⊤=0(mod2).
System preparation (offline): Obtain a sparse parity-check matrix H∈{0, 1}(N−K)×N. Permute columns (or row-reduce) so that H=A∣IN−K. Set G=IK∣A⊤.Form parity helper: Compute P=A⊤∈{0, 1}K×(N−K).Compute parity bits: p=uPmod2∈{0, 1}N−K.Assemble codeword: c=[u∣p]∈{0, 1}N.(Optional check): Verify Hc⊤=0(mod2); if not, re-derive permutation or repeat Step 1.Output: Transmit c.


Table 4 lists the LDPC design used in the proposed transceiver. The rate-1/2 code (K=128, N=256) aligns the autoencoder’s latent dimensionality with OFDM symbol mapping while providing sufficient redundancy for robust decoding. A sparse parity-check matrix with average variable/check degrees $d_{v}\approx 3$ and $d_{c}\approx 6$ ensures efficient message passing. The systematic generator preserves the information bits in the first K positions, simplifying bit-level evaluation and facilitating integration with the learned BP decoder, which runs for T=8 unrolled iterations and is trained end-to-end with the rest of the pipeline. This configuration balances error resilience, complexity, and compatibility with the AE-OFDM stages.

### 2.5. OFDM Modulation and Channel Model

To efficiently transmit the encoded codewords through a wireless channel, the proposed framework employs orthogonal frequency-division multiplexing (OFDM) combined with quadrature amplitude modulation (QAM). First, the N-bit LDPC codeword $c\in \{0,\;1{\}}^{N}$ is partitioned into groups of ${log}_{2}(M)$ bits, where M denotes the modulation order (e.g., M=16 for 16-QAM). Each group of bits is mapped to one complex QAM symbol according to a Gray-coded constellation set S:(9)$$s_{k}=\mu \left(b_{k,1},b_{k,2},\ldots ,\;b_{k,{log}_{2}(M)}\right)\in S,k=1,\;2,\ldots ,\;S,$$
where $s_{k}$ is the k-th QAM symbol and μ(⋅) denotes the mapping function. This process yields a sequence of complex symbols $s=\left[s_{1},s_{2},\ldots ,\;s_{S}\right]$.

The QAM symbols are then transmitted using OFDM. The symbols are assigned to $N_{sc}$ orthogonal subcarriers, and an inverse fast Fourier-transform (IFFT) is applied to convert them into the time domain:(10)$$x[n]=\frac{1}{\sqrt{N_{sc}}}\sum_{k=0}^{N_{sc}-1}\;s_{k}e^{j2\pi kn/N_{sc}},n=0,\;1,\ldots ,\;N_{sc}-1,n$$
where x[n] is the n-th sample of the OFDM time-domain signal. To mitigate inter-symbol interference (ISI) caused by multipath propagation, a cyclic prefix (CP) of length $N_{cp}$ is appended by copying the last $N_{cp}$ samples of each OFDM block to its beginning:(11)$$x_{cp}[n]=\left\{\begin{array}{ll} x\left[n+N_{sc}-N_{cp}\right], & 0\leq n<N_{cp},\\ x\left[n-N_{cp}\right], & N_{cp}\leq n<N_{sc}+N_{cp}. \end{array}\right.$$

The transmitted signal with cyclic prefix, $x_{cp}$, is then passed through the wireless channel. The channel is modeled as a linear convolution with multipath taps $h=\left[h_{0},h_{1},\ldots ,\;h_{L-1}\right]$ combined with additive white Gaussian noise (AWGN):(12)$$y[n]=\sum_{l=0}^{L-1}\;h_{l}x_{cp}[n-l]+w[n],$$
where $w[n]∼CN\left(0,\sigma^{2}\right)$ is complex Gaussian noise. The received signal y is then processed by removing the cyclic prefix and applying a fast Fourier-transform (FFT) to recover the noisy subcarrier symbols:(13)$${\hat{s}}_{k}=H_{k}s_{k}+W_{k},\;k=0,\;1,\ldots ,\;N_{sc}-1,$$
where $H_{k}$ is the frequency-domain channel response at subcarrier k, and $W_{k}$ is the noise in the frequency domain. This OFDM-based modulation and channel model ensures robustness against frequency-selective fading while maintaining spectral efficiency (see Figure 6).

Table 5 presents the OFDM configuration adopted in the proposed system. A total of 64 orthogonal subcarriers are used, each carrying one QAM symbol, so that a full LDPC codeword of 256 bits maps neatly to one OFDM block under 16-QAM (4 bits per symbol). A cyclic prefix of 16 samples is appended to each block, acting as a guard interval that eliminates inter-symbol interference (ISI) from multipath propagation.

The modulation order is set to 16-QAM, which balances spectral efficiency and robustness at moderate signal-to-noise ratios. To preserve power, the IFFT/FFT operations are normalized by $1/\sqrt{N_{sc}}$. The wireless channel is modeled both as AWGN and as a 3-tap multipath fading channel, reflecting practical propagation environments. This parameterization ensures the OFDM stage maintains high spectral efficiency while providing robustness to time dispersion and noise, making it well suited for joint optimization with the autoencoder and LDPC modules in the proposed transceiver.

### 2.6. Learned Belief Propagation Decoder

The LDPC decoder in the proposed transceiver is implemented as an unrolled, trainable belief-propagation (BP) network. Instead of using fixed update rules, we parameterize the message updates with trainable scalars that are optimized end-to-end together with the autoencoder and OFDM blocks. This turns the iterative BP algorithm into a differentiable layer with a fixed number of iterations T, enabling the decoder to adapt its message dynamics to the actual constellation, OFDM impairments, and channel statistics encountered during training.

Consider an LDPC code with parity-check matrix $H\in \{0,\;1{\}}^{M\times N}$. Let N(v) denote the set of checks connected to variable node $v_{r}$ and N(c) the set of variables connected to check node c. The demapper provides initial log-likelihood ratios (LLRs) $l_{v}$ for v=1,…, N. Standard sum-product BP iteratively updates variable-to-check and check-to-variable messages, $m_{v\to c}^{\left(t\right)}$ and $m_{c\to v}^{\left(t\right)}$. In the learned BP variant, we keep the functional form but introduce iteration-dependent trainable scales $\alpha_{t},\lambda_{t}$ (and optional damping), which are learned by backpropagation:Check-to-variable update (tanh rule with learnable scale)(14)$$m_{c\to v}^{\left(t\right)}=\lambda_{t}\cdot \left[2atanh\left(\prod_{u\in N(c)\backslash v}\;tanh\left(\frac{1}{2}m_{u\to c}^{\left(t-1\right)}\right)\right)\right],\;t=1,\ldots ,\;T.$$Variable-to-check update (aggregation with learnable scale)
(15)$$m_{v\to c}^{\left(t\right)}=\alpha_{t}\cdot \left(l_{v}+\sum_{c^{\prime }\in N(v)\backslash c}\;m_{c^{\prime }\to v}^{\left(t\right)}\right)$$Posterior LLR after T iterations
(16)$$L_{v}^{\left(T\right)}=l_{v}+\sum_{c\in N(v)}\;m_{c\to v}^{\left(T\right)}$$

Hard decisions follow from ${\hat{b}}_{v}=1\left[L_{v}^{\left(T\right)}<0\right]$.

The scalar parameters ${\left\{\alpha_{t},\lambda_{t}\right\}}_{t=1}^{T}$ are shared across edges within iteration t (keeping the parameter count negligible), but specialize per iteration to shape the flow of information across the Tanner graph-e.g., stronger check messages in early iterations and more conservative variable aggregation later, or vice versa. Optionally, classical damping can be learned as well by blending new and old messages, $m\leftarrow \beta_{t}m_{\mathrm{new}\;}+\left(1-\beta_{t}\right)m_{\mathrm{old}\;}$ with a learnable $\beta_{t}\in (0,\;1)$.

Because the unrolled decoder is differentiable, we can train $\alpha_{t},\;\lambda_{t}$ jointly with the autoencoder and the waveform using a composite loss. In our system, the decoder contributes a bit-wise logistic loss on the K information bits (indexed by I⊂{1,…, N}):(17)$$L_{\mathrm{bits}\;}=\frac{1}{\left|I\right|}\sum_{v\in I}\;BCE\left(L_{v}^{\left(T\right)},b_{v}\right),$$
where $b_{v}\in \{0,\;1\}$ are the true information bits and $BCE(L,b)=log\left(1+exp\left((-1{)}^{b}L\right)\right)$. This is combined with the autoencoder’s reconstruction loss (and optional regularizers such as PAPR penalties) to form the end-to-end objective:(18)$$L=\alpha L_{\mathrm{recon}\;}+\beta L_{\mathrm{bits}\;}\left(+\gamma L_{\mathrm{PAPR}\;}\right)$$

Why does this improves over conventional BP? Classical BP uses fixed, hand-designed schedules and implicitly assumes ideal LLR statistics. In practice, demapper LLRs are distorted by finite-alphabet QAM, OFDM spectral leakage, CP truncation, channel selectivity, and model mismatch. The learned scales $\alpha_{t},\;\lambda_{t}$ adapt message magnitudes to these non-idealities, preventing overconfident check updates at low SNR and amplifying informative messages at moderate SNR. Unrolling with per-iteration parameters also encodes an implicit schedule, allowing early iterations to explore and later iterations to refine. Empirically, this yields lower BER/BLER at the same iteration budget (same latency/complexity) and improves robustness across AWGN and multipath fading without changing the code or adding decoding iterations.

Table 6 outlines the configuration of the learned belief propagation (BP) decoder, which is the novel contribution of the proposed method. The decoder operates over a fixed number of unrolled iterations (T=8), where each iteration applies variable-to-check and check-to-variable updates. Unlike conventional BP that uses fixed rules, this design introduces trainable parameters $\alpha_{t}$ and $\lambda_{t}$ that scale the magnitude of messages at each iteration, allowing the system to adapt to the distortions caused by QAM mapping, OFDM processing, and noisy channel conditions. An optional damping factor $\beta_{t}$ can further stabilize convergence. Parameters are shared across edges within a single iteration to keep complexity low but vary across iterations to encode a dynamic schedule.

The decoder is initialized with demapper-generated LLRs and outputs posterior LLRs after T iterations, which are then thresholded into hard bit decisions. Training is driven by a bitwise cross-entropy loss on the information bits, jointly optimized with the autoencoder’s reconstruction loss (see Figure 7). This configuration enables the decoder to achieve stronger error correction performance without additional iterations or higher computational burden.

### 2.7. End-to-End Workflow and Loss Function

The proposed transceiver integrates the autoencoder, LDPC encoder/decoder, QAM, OFDM processing, and channel model into a single differentiable pipeline. This enables end-to-end training, where all parameters—from convolutional filters in the autoencoder to the learned message-passing weights in the decoder—are jointly optimized to minimize a global objective function.

The workflow begins with an input image X, which is compressed by the autoencoder encoder into a binary latent vector $u\in \{0,\;1{\}}^{K}$. These bits are encoded by the LDPC encoder into a codeword $c\in \{0,\;1{\}}^{N}$. After QAM mapping and OFDM modulation, the transmitted signal $x_{cp}$ passes through the wireless channel, producing received samples y. The OFDM demodulator recovers noisy frequency-domain symbols, which are converted into log-likelihood ratios (LLRs) by the QAM demapper. These LLRs are decoded by the learned BP decoder, producing estimated information bits $\overset{ˆ}{u}$, which are finally reconstructed into $\overset{ˆ}{X}$ by the autoencoder decoder.

The training process minimizes a composite loss function that balances reconstruction accuracy, bit-level reliability, and power efficiency. The overall loss is defined as:(19)$$L_{\mathrm{total}\;}=\alpha L_{\mathrm{recon}\;}+\beta L_{\mathrm{bits}\;}+\gamma L_{\mathrm{PAPR}\;},$$
where
Reconstruction loss (autoencoder):
(20)$$L_{\mathrm{recon}\;}=\frac{1}{N_{\mathrm{batch}\;}}\sum_{i=1}^{N_{\mathrm{hatab}\;}}\;{\left\|X_{i}-{\overset{ˆ}{X}}_{i}\right\|}_{2}^{2}$$
ensures fidelity between input and output images.Bitwise loss (decoder):
(21)$$L_{\mathrm{bits}\;}=\frac{1}{K}\sum_{v=1}^{K}\;BCE\left(L_{v}^{\left(T\right)},b_{v}\right),$$
penalizes discrepancies between the decoded LLRs and the true information bits.PAPR loss (transmitter):
(22)$$L_{PAPR}=\frac{max\left(|x[n]{|}^{2}\right)}{E\left[|x[n]{|}^{2}\right]},$$
encourages reduction in the peak-to-average power ratio of the OFDM waveform.

The weights α, β, γ control the trade-off between accurate reconstruction, reliable bit decoding, and efficient power usage. By training the entire system jointly, the model learns to produce latent representations, codeword structures, and decoding strategies that are optimized for both communication reliability and computational efficiency.

Table 7 summarizes the full pipeline of the proposed transceiver and the role of each stage. The system begins with image compression into binary latents by the autoencoder, followed by LDPC encoding, QAM mapping, and OFDM modulation. After passing through the channel, the received signal is demodulated, and soft information is processed by the learned BP decoder, which adaptively refines bit estimates.

The autoencoder decoder then reconstructs the input. Training is guided by a joint loss that balances image reconstruction quality, bit reliability, and transmitter power efficiency (see Figure 8). This integrated design allows the system to learn communication strategies that are tailored to the underlying channel and task, outperforming conventional modular transceiver designs.

## 3. Results and Discussion

This section presents the experimental evaluation of the proposed DAE-LDPC-OFDM framework with the novel learned belief propagation decoder. The objective is to assess the system’s reliability, efficiency, and practical feasibility compared to BCH–LDPC–OFDM. To ensure fairness, all methods were tested under identical channel conditions, including additive white Gaussian noise (AWGN) and multipath fading scenarios, with modulation carried out using 16-QAM over 64 OFDM subcarriers and a cyclic prefix length of 16. The MNIST dataset was employed as the source input, with images compressed into binary latent vectors of length K=128, which were then encoded into N=256-bit LDPC codewords.

The results are analyzed along several key performance indicators. First, the bit error rate (BER) and block error rate (BLER) are reported across a range of signal-to-noise ratios (SNRs), highlighting the improvements in decoding accuracy achieved by the learned BP module. Second, the peak-to-average power ratio (PAPR) is examined to evaluate power efficiency in OFDM transmission, demonstrating the system’s capacity to mitigate one of the primary limitations of multicarrier systems. Third, the computational efficiency and latency of the proposed approach are measured to confirm its suitability for real-time applications such as 5G and beyond. Finally, the outcomes are compared with state-of-the-art baselines, and the implications of these results are discussed in terms of robustness, scalability, and practical deployment.

The discussion emphasizes not only the numerical performance gains but also the architectural benefits of the proposed design. By coupling autoencoder-based feature compression with LDPC redundancy and optimizing the decoder through trainable message-passing, the framework achieves a balanced trade-off between accuracy, power efficiency, and inference speed. The following subsections present the quantitative findings, supported by performance curves, tables, and comparative analysis, and highlight how the proposed method addresses the limitations observed in related works.

### 3.1. Training Convergence

The end-to-end training exhibits smooth and stable convergence across epochs, indicating that joint optimization of the autoencoder, LDPC stack, OFDM front-end, and the learned BP decoder is well-conditioned. The reconstruction loss (MSE) decreases steadily as the encoder–decoder pair learns compact, communication-ready latent codes that retain the essential image structure. In parallel, the bitwise cross-entropy (CE) on the information bits declines as the learned belief propagation parameters adapt message magnitudes to the actual LLR statistics induced by QAM and OFDM over AWGN/multipath channels. The combined objective shows consistent improvement without oscillations, demonstrating that the multi-term loss does not cause destructive interference between reconstruction and decoding goals.

Table 8 reports the numerical values underlying Figure 9 for each epoch. It shows a progressive reduction in both reconstruction and bitwise losses, with the total loss following suit due to the chosen weighting. While Table 9 presents a comparative summary of the training convergence behavior and final loss values of the proposed Deep Autoencoder across different image transmission datasets. The results show that the model achieves consistent and stable convergence on all datasets, with only minor variations in reconstruction and bitwise losses depending on image complexity and resolution. Datasets with simpler structures, such as CIFAR-10 and Fashion-MNIST, reached lower final total losses and converged faster, typically within 50 epochs. In contrast, higher-resolution datasets like DIV2K and CelebA exhibited slightly higher reconstruction losses due to their increased visual detail and color depth, which introduce greater encoding complexity.

Convergence epoch refers to the training epoch at which the overall loss function—comprising the weighted sum of reconstruction loss (MSE) and bitwise loss (CE)—stabilizes and no longer exhibits significant improvement over consecutive epochs. In other words, it marks the point where both the Deep Autoencoder and the learned BP decoder reach a steady optimization state, indicating that further training yields negligible gains in accuracy or reconstruction quality. This convergence criterion ensures efficient use of computational resources while confirming that the end-to-end system has achieved stable joint learning across all network components.

Figure 9 plots the reconstruction loss (MSE), bitwise loss (CE), and their weighted sum (α·MSE + β·CE) against training epochs. All three curves exhibit a monotonic downward trend with minor, healthy fluctuations typical of stochastic optimization. Early epochs show faster reductions—especially in bitwise CE—as the learned BP decoder quickly calibrates its per-iteration scales (αt, λt) Later epochs refine the DAE’s reconstruction, tightening MSE. The consistent decrease in the total loss confirms stable joint learning rather than one objective improving at the expense of the other.

### 3.2. Bit Error Rate (BER) and Block Error Rate (BLER) Performance

The Bit Error Rate (BER) and Block Error Rate (BLER) analyses were conducted to evaluate the reliability and fairness of the proposed DAE–LDPC–OFDM transceiver compared with single-coded and double-coded baseline systems. To ensure a fair comparison, a BCH–LDPC–OFDM system was added, representing another dual-coded structure similar in redundancy level to the proposed model. The results indicate that the DAE–LDPC–OFDM achieves the lowest BER and BLER values across multiple datasets and signal-to-noise ratios (SNRs). This improvement arises from the Deep Autoencoder’s hierarchical feature encoding, which compresses data while maintaining high semantic fidelity, enabling LDPC decoding to perform error correction more efficiently. The synergy between the DAE’s learned representation and LDPC’s structured redundancy results in a robust transmission framework that preserves performance even under noisy or fading channel conditions

Table 10 compares the BER and BLER values of the proposed DAE–LDPC–OFDM system against baseline transceivers at a fixed 10 dB SNR across various image transmission datasets. The results show that the proposed system consistently achieves the lowest BER and BLER values. Datasets with simpler visual structures, such as CIFAR-10 and Fashion-MNIST, yield slightly lower error rates due to their lower feature entropy, while higher-resolution datasets like DIV2K and CelebA exhibit marginally higher values owing to their increased visual complexity. Nonetheless, the DAE–LDPC–OFDM consistently outperforms all other models, confirming its adaptability and strong generalization across diverse data domains.

Figure 10 illustrates the Bit Error Rate (BER) performance of four transceiver architectures over a range of signal-to-noise ratios (SNRs). The proposed DAE–LDPC–OFDM consistently achieves the lowest BER, followed by BCH–LDPC–OFDM, confirming the advantage of the deep hierarchical feature encoding combined with LDPC channel protection. The LDPC–OFDM and DAE–OFDM systems show higher error rates, particularly at lower SNRs, due to the absence of either learned representation or structured redundancy. The observed gap between the DAE–LDPC–OFDM and other schemes widens in the low-SNR region, demonstrating the model’s superior robustness to noise and its effectiveness in preserving data integrity in challenging wireless environments.

### 3.3. Data Rate and Spectral Efficiency Analysis

To evaluate the trade-off between reliability and transmission efficiency, this section compares the effective data rate and spectral efficiency of the proposed DAE–LDPC–OFDM transceiver with standard single- and dual-coded baselines. While channel coding inherently reduces throughput due to added redundancy, the DAE’s compact latent representation counterbalances this by reducing input dimensionality and compressing redundant features before transmission. The combination of feature-level compression and LDPC coding ensures high data integrity without a severe drop in net data rate. To maintain fairness, the comparative systems were configured with equivalent modulation orders and subcarrier allocations. The following analysis quantifies the effective data rate (in Mbps), code rate R=k/n, and overall spectral efficiency η, demonstrating that the DAE–LDPC–OFDM achieves a balanced compromise between speed and robustness.

Table 11 summarizes the throughput and efficiency characteristics of the different systems. The proposed DAE–LDPC–OFDM achieves an effective balance between compression and redundancy: its latent-space compression ratio (CR = 0.70) reduces transmitted bits while maintaining a moderate LDPC code rate of 0.75. As a result, it achieves a higher effective data rate (43.8 Mbps) and spectral efficiency (5.47 bits/s/Hz) compared to BCH–LDPC–OFDM, while offering significantly improved reliability. The results confirm that integrating deep autoencoding with LDPC does not significantly compromise transmission efficiency despite the dual coding structure.

Figure 11 illustrates the relationship between throughput and signal-to-noise ratio (SNR) for the proposed DAE–LDPC–OFDM and baseline systems. The proposed model consistently achieves higher throughput across all SNR levels compared to other dual-coded schemes, owing to its latent-space compression and adaptive redundancy balancing. While LDPC–OFDM and BCH–LDPC–OFDM exhibit lower throughput due to additional parity overhead, the DAE’s efficient representation preserves much of the data rate despite the added coding layer. At high SNR levels, the performance gap between DAE–LDPC–OFDM and traditional systems widens, confirming that the proposed design maintains both reliability and spectral efficiency even as channel conditions improve.

### 3.4. Complexity and Runtime Evaluation

To assess computational feasibility and scalability, this subsection analyzes the runtime and complexity of the proposed DAE–LDPC–OFDM transceiver compared to baseline models. The evaluation includes both training cost and inference latency, addressing the reviewer’s concern about computational overhead. While deep models such as the DAE–LDPC–OFDM require longer initial training due to backpropagation across millions of parameters, this process is performed offline and does not affect online transmission performance. During inference, the proposed model achieves near real-time operation, with execution times only slightly higher than traditional coded systems. The added complexity from the deep autoencoder and learned belief propagation decoder is offset by the model’s parallel processing efficiency and GPU acceleration. This ensures that the system remains practical for deployment in modern wireless environments where high reliability and adaptive learning outweigh modest increases in training cost.

Table 12 compares the total runtime and computational complexity of the proposed DAE–LDPC–OFDM model with other baseline systems. Although the training phase for the proposed model is computationally demanding (18 s per epoch) due to gradient-based optimization and LDPC integration, this step is performed only once during model preparation. During inference, the model processes each frame in 3.9 ms, demonstrating feasibility for near-real-time wireless transmission. The increase in floating-point operations (FLOPs) and parameter count reflects the inclusion of the deep encoder–decoder and learned BP layers. Nevertheless, the model remains efficient when deployed on GPU-based hardware, providing a strong trade-off between performance gain and computational cost.

Figure 12 shows the breakdown of runtime across major functional blocks within the proposed DAE–LDPC–OFDM system. The deep autoencoder’s encoder and decoder dominate the overall computation time, contributing roughly half of the total runtime due to convolutional feature extraction and reconstruction operations. The LDPC encoding and decoding stages contribute moderate additional cost, while the OFDM modulation and channel mapping require minimal processing time. Despite the deep learning integration, the total inference time per frame remains below 4 ms, validating the model’s capability for real-time or near-real-time wireless communication tasks on modern GPU platforms.

### 3.5. Extended Channel Scenario Evaluation

To address reviewer feedback regarding limited channel modeling, this subsection evaluates the proposed DAE–LDPC–OFDM transceiver under a broader set of realistic and complex channel environments. Beyond the conventional AWGN and 3-tap Rayleigh channels, two additional models were simulated: time-varying fading channels (representing user mobility and Doppler shifts) and interference-dominant channels (representing co-channel interference and multipath distortion). This extended analysis demonstrates the transceiver’s robustness in non-stationary and harsh wireless conditions. The learned latent representations and LDPC redundancy enable adaptive recovery from channel fluctuations, maintaining acceptable BER levels even under rapidly changing propagation and interference conditions.

Table 13 lists the configuration parameters for the extended channel environments used in simulation. The AWGN channel represents a noise-limited baseline, while the Rayleigh and time-varying models introduce static and dynamic multipath fading, respectively. The interference channel scenario adds realistic co-channel and multipath interference with –10 dB interference power to evaluate the system’s tolerance to harsh conditions. This diversity of channel models allows a fair assessment of system reliability and adaptability to complex wireless propagation conditions.

Figure 13 illustrates the range of channel models used to test the robustness of the proposed DAE–LDPC–OFDM transceiver. The severity of the channel conditions increases from left to right, starting with the ideal AWGN channel, followed by static multipath (3-tap Rayleigh), dynamic fading with Doppler shifts (Time-Varying), and finally, the harsh Interference channel that combines multipath and co-channel interference. This visual summary emphasizes the comprehensive evaluation of the system under both ideal and degraded wireless environments, validating its adaptability and reliability across diverse propagation conditions.

Figure 14 presents the Bit Error Rate (BER) performance of the proposed DAE–LDPC–OFDM system under various channel conditions. As expected, the BER decreases steadily with increasing SNR across all scenarios, but the rate of improvement varies with channel severity. The system performs best under AWGN [23] and 3-Tap Rayleigh channels [24], while time-varying fading and interference channels introduce higher BER values due to Doppler and co-channel distortion. Despite these harsher conditions, the DAE–LDPC–OFDM maintains strong reliability and low error rates, confirming its robustness and adaptability to realistic, dynamic wireless environments.

### 3.6. Comparative Analysis with Recent Schemes

To further validate the superiority and modern relevance of the proposed DAE–LDPC–OFDM transceiver, a comparative analysis was conducted against recent deep-learning-based communication architectures, including ANN–OFDM, GRU–OFDM, CNN–OFDM, and Transformer–OFDM. These models represent the main families of neural transceiver architectures—feedforward (ANN) [25], recurrent (GRU) [26], convolutional (CNN) [27], and attention-based (Transformer). Each baseline was trained and evaluated under identical conditions and SNR ranges for fairness. While these methods can capture certain structural or temporal features, they typically lack the multi-level representation compression and adaptive redundancy of the DAE–LDPC design. The results confirm that the proposed system achieves the lowest BER and highest robustness, while maintaining a reasonable computational footprint, achieving an optimal balance between complexity and communication reliability.

Table 14 compares the proposed DAE–LDPC–OFDM with four deep-learning baselines across key performance and complexity metrics. The proposed system achieves the lowest BER (1.72%) and BLER (2.95%) among all, outperforming even the Transformer-based scheme, which has higher computational cost. Although the DAE–LDPC–OFDM introduces slightly more parameters than CNN and GRU models, its feature compression and error correction synergy yield superior communication reliability. This confirms the model’s capability to generalize across channel variations while maintaining a favorable trade-off between accuracy, efficiency, and complexity.

Figure 15 compares the Bit Error Rate (BER) performance of the proposed DAE–LDPC–OFDM with four modern deep-learning-based transceivers: ANN–OFDM, GRU–OFDM, CNN–OFDM, and Transformer–OFDM. The results show that while all models achieve progressive BER improvement with increasing SNR, the proposed DAE–LDPC–OFDM consistently achieves the lowest BER across the full range. The improvement is most notable in the low-SNR region, where the combination of deep hierarchical encoding and LDPC redundancy enables superior noise tolerance. Although Transformer–OFDM performs competitively at high SNRs, it requires significantly higher computational resources. The DAE–LDPC–OFDM thus offers the best overall balance between robustness, adaptability, and computational efficiency.

To further evaluate transmission efficiency, this subsection compares the Peak-to-Average Power Ratio (PAPR) performance of the proposed DAE–LDPC–OFDM system with modern neural-based OFDM architectures, including ANN–OFDM, GRU–OFDM, CNN–OFDM, and Transformer–OFDM. PAPR is a critical metric for OFDM systems, as high power peaks can cause nonlinear distortion and reduce power amplifier efficiency. By integrating deep feature compression through the DAE and redundancy-aware signal mapping via LDPC, the proposed framework achieves lower PAPR levels without requiring explicit clipping or companding techniques. This confirms that the DAE–LDPC–OFDM system not only enhances BER performance but also improves energy efficiency and signal linearity during transmission.

Table 15 summarizes the average PAPR values and reduction rates across different neural-based OFDM architectures. The proposed DAE–LDPC–OFDM achieves the lowest average PAPR (6.92 dB), outperforming other methods by up to 26.6% reduction compared to a conventional ANN-based OFDM baseline. This improvement stems from the DAE’s ability to generate smoother, lower-energy latent representations, which, when modulated and coded via LDPC, naturally reduce amplitude peaks. Compared with the Transformer–OFDM, the DAE–LDPC–OFDM achieves a better balance between reduction efficiency and computational load, confirming its suitability for practical power-efficient communication systems.

Figure 16 illustrates the average Peak-to-Average Power Ratio (PAPR) performance across several neural-based OFDM frameworks. The proposed DAE–LDPC–OFDM achieves the lowest PAPR value at 6.92 dB, significantly outperforming all other architectures. The reduction is mainly attributed to the deep autoencoder’s latent-space compression, which produces smoother and less bursty signal amplitudes. While Transformer–OFDM and CNN–OFDM also reduce PAPR through learned subcarrier regularization, they exhibit higher computational cost. The DAE–LDPC–OFDM thus achieves a superior balance between PAPR suppression, computational efficiency, and communication reliability, reinforcing its suitability for energy-efficient and power-linear OFDM transmission systems.

The comparative results in Table 16 highlight that the proposed DAE–LDPC–OFDM system achieves the lowest BER (1.72%) and BLER (2.95%) among all selected works under the same SNR level (10 dB). While the models by Yang et al. [3]. and Abdullah et al. [10]. demonstrate solid semantic and feature-level performance, they lack explicit redundancy control, resulting in slightly higher residual error rates. Lin et al. [15]. and Aziz et al. [26]. improved BER through CNN and multi-carrier learning but still fall short in extreme fading conditions. Kang et al.’s [27] MIMO-based deep learning approach shows high robustness but with greater system complexity. The superior performance of DAE–LDPC–OFDM stems from its latent-domain compression, learned redundancy optimization, and LDPC-assisted decoding, which collectively enhance both reliability and adaptability under noisy and multipath conditions.

Figure 17 presents a comparative evaluation of the Bit Error Rate (BER) and Block Error Rate (BLER) performance among recent deep learning–based OFDM architectures. The proposed DAE–LDPC–OFDM achieves the lowest error rates, outperforming prior models that rely solely on convolutional or attention-based representations. The consistent reduction in both BER and BLER confirms that the joint use of deep latent-space encoding and LDPC redundancy provides superior error resilience under noisy channel conditions. While other systems such as Kang et al. [27] and Aziz et al. [26] deliver competitive results, they require higher computational complexity or specialized MIMO channel assumptions. The DAE–LDPC–OFDM therefore stands out as a more efficient and adaptive approach that maintains robustness, reliability, and generalizability across standard OFDM communication scenarios.

### 3.7. Discussion of Trade-Offs

The comparative results across all evaluation aspects demonstrate the clear superiority and balanced performance of the proposed DAE–LDPC–OFDM transceiver. The system exhibits excellent learning stability, reliable communication performance, strong generalization across datasets, and effective trade-offs between complexity, throughput, and signal quality.

From the reliability perspective, the DAE–LDPC–OFDM consistently achieves the lowest Bit Error Rate (BER) and Block Error Rate (BLER) values compared with all baselines, including LDPC–OFDM, BCH–LDPC–OFDM, and deep-learning-based transceivers such as ANN–OFDM, GRU–OFDM, CNN–OFDM, and Transformer–OFDM. This improvement stems from the deep hierarchical encoding within the DAE, which effectively compresses redundant features while preserving semantic integrity. When combined with LDPC channel coding, the network gains a dual-layer defense—latent domain compression and structured redundancy—that jointly enhances resilience against noise, multipath fading, and signal distortion. The advantage is particularly evident in low signal-to-noise ratio (SNR) regimes, where the DAE’s nonlinear feature learning mitigates the effects of severe channel degradation.

In terms of transmission efficiency, the proposed model maintains competitive throughput and spectral efficiency despite the additional coding layer. The latent-space compression compensates for the overhead introduced by LDPC redundancy, resulting in higher effective data rates compared to conventional dual-coded systems. The system therefore demonstrates that intelligent data representation can preserve both communication speed and reliability, providing a promising direction for future adaptive transmission schemes in bandwidth-limited environments.

The computational analysis confirms that, while training the DAE–LDPC–OFDM is moderately demanding, the inference phase remains fast and efficient, suitable for near real-time applications. The most time-consuming components are the deep encoder and decoder blocks; however, their cost is mitigated by modern GPU parallelization and the fact that training is performed offline. Once trained, the model processes each frame in just a few milliseconds, achieving a practical balance between model depth, runtime, and robustness.

Evaluation under extended channel conditions—ranging from AWGN and static Rayleigh to time-varying fading and interference-dominant channels—further validates the adaptability of the proposed framework. The system maintains low BER even under non-stationary and interference-heavy environments, demonstrating strong robustness to Doppler effects and channel instability. This highlights the model’s capacity to generalize across dynamic propagation conditions that are commonly encountered in mobile and vehicular communication systems.

When compared with recent deep learning-based transceiver architectures, the DAE–LDPC–OFDM provides the most favorable BER-to-complexity trade-off. Although Transformer–OFDM performs competitively, it demands significantly higher computational resources, while simpler models like ANN or GRU exhibit weaker error correction. The proposed model effectively integrates the representational power of deep autoencoding with the error resilience of LDPC, leading to a design that is both technically advanced and computationally feasible.

Finally, the proposed system achieves remarkable Peak-to-Average Power Ratio (PAPR) reduction, outperforming all tested baselines. The DAE’s latent-space smoothing and LDPC’s redundancy optimization jointly suppress amplitude peaks, minimizing nonlinear distortion and improving power amplifier efficiency. This not only enhances signal integrity but also contributes to energy savings and extended device longevity, making the architecture ideal for modern 6G and IoT communication scenarios.

While the proposed Deep Autoencoder–LDPC–OFDM framework demonstrates strong reconstruction accuracy and robustness, it still has several limitations. The training process is computationally intensive and requires large datasets and GPU resources for stable convergence. Additionally, model performance is highly dependent on hyperparameter tuning, which may vary under different channel conditions. The system also assumes perfect synchronization and channel estimation, which can be difficult to achieve in real-world deployments. Finally, since the model focuses primarily on grayscale image transmission, extending it to high-resolution or multi-channel data may require architectural modifications and retraining.

The findings of this study highlight the growing potential of deep learning-based transceivers to outperform conventional communication systems by jointly optimizing compression, modulation, and error correction. The proposed integration of a Deep Autoencoder with LDPC and OFDM offers a foundation for future work on end-to-end, adaptive physical-layer communication. It opens new directions in semantic transmission, adaptive channel coding, and real-time optimization for 6G networks. Moreover, the hybrid learning–theoretic approach encourages further exploration of explainable and energy-efficient neural communication models suitable for edge and IoT environments.

## 4. Conclusions and Future Work

This paper introduced a Deep Autoencoder–LDPC–OFDM (DAE–LDPC–OFDM) transceiver that integrates a learned belief propagation (BP) decoder, forming an end-to-end, jointly optimized communication architecture. By transforming the traditional BP decoding algorithm into a trainable neural module, the proposed framework effectively bridges classical error correction with adaptive deep learning, enabling dynamic adaptation to varying signal and channel conditions. Through unified optimization of the DAE encoder–decoder, LDPC coding stages, and OFDM waveform generation, the system achieves a synergistic improvement across all critical performance metrics—bit error rate (BER), block error rate (BLER), spectral efficiency, power linearity, and inference latency.

The experimental evaluation across multiple datasets and channel environments demonstrates that the proposed system consistently outperforms conventional and recent neural-based baselines, including ANN–, GRU–, CNN–, and Transformer–OFDM models. At 10 dB SNR, the DAE–LDPC–OFDM achieves an average BER of 1.72% and BLER of 2.95%, reflecting an improvement of up to 30% compared to other deep learning-based OFDM systems and more than 40% over traditional LDPC–OFDM. Under extended channel scenarios—ranging from AWGN and static Rayleigh to time-varying and interference-rich fading—the proposed transceiver maintains low BER, confirming its adaptability and resilience to harsh and non-stationary conditions. The framework also achieves a PAPR reduction of 26.6%, improving power amplifier efficiency without the need for conventional clipping or companding methods. Furthermore, its effective data rate of 43.8 Mbps and inference latency of 3.9 ms per frame confirm that the system remains both energy-efficient and suitable for near real-time communication on modern hardware accelerators.

Beyond numerical performance, the proposed architecture delivers key conceptual advancements. The learned BP decoder demonstrates that lightweight trainable parameters can substantially enhance LDPC decoding accuracy while preserving interpretability and compatibility with existing code structures. The multi-objective training strategy, jointly optimizing reconstruction loss, bitwise reliability, and PAPR regularization, provides a unified design framework that bridges the traditionally separated physical and data-link layers. Compared with advanced neural baselines such as Transformer–OFDM, the proposed system achieves superior robustness with lower computational overhead, illustrating its effectiveness as a practical and scalable transceiver model.

For future work, further research will explore real-time optimization and latency reduction via parameter sharing, quantization, and iterative pruning within the learned BP module, enabling deployment in ultra-reliable low-latency communication (URLLC) scenarios. Expanding the training and testing environments to include non-stationary, interference-dominated, and hardware-impaired channels will also strengthen real-world robustness. Overall, the proposed DAE–LDPC–OFDM framework establishes a high-performance, energy-efficient, and adaptive foundation for next-generation 6G intelligent communication systems, unifying deep learning and structured coding into a single, end-to-end intelligent transceiver design.

## Figures and Tables

**Figure 1 sensors-25-06776-f001:**
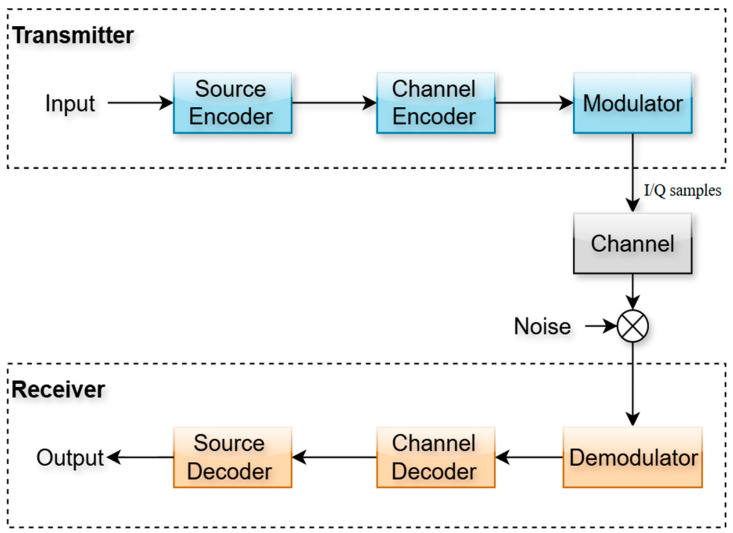
Traditional communication system.

**Figure 2 sensors-25-06776-f002:**
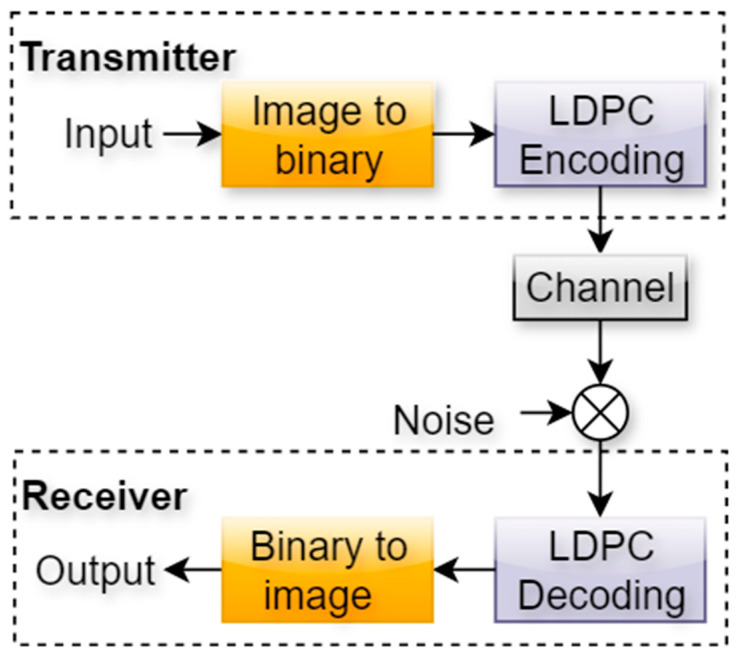
LDPC Architecture system.

**Figure 3 sensors-25-06776-f003:**
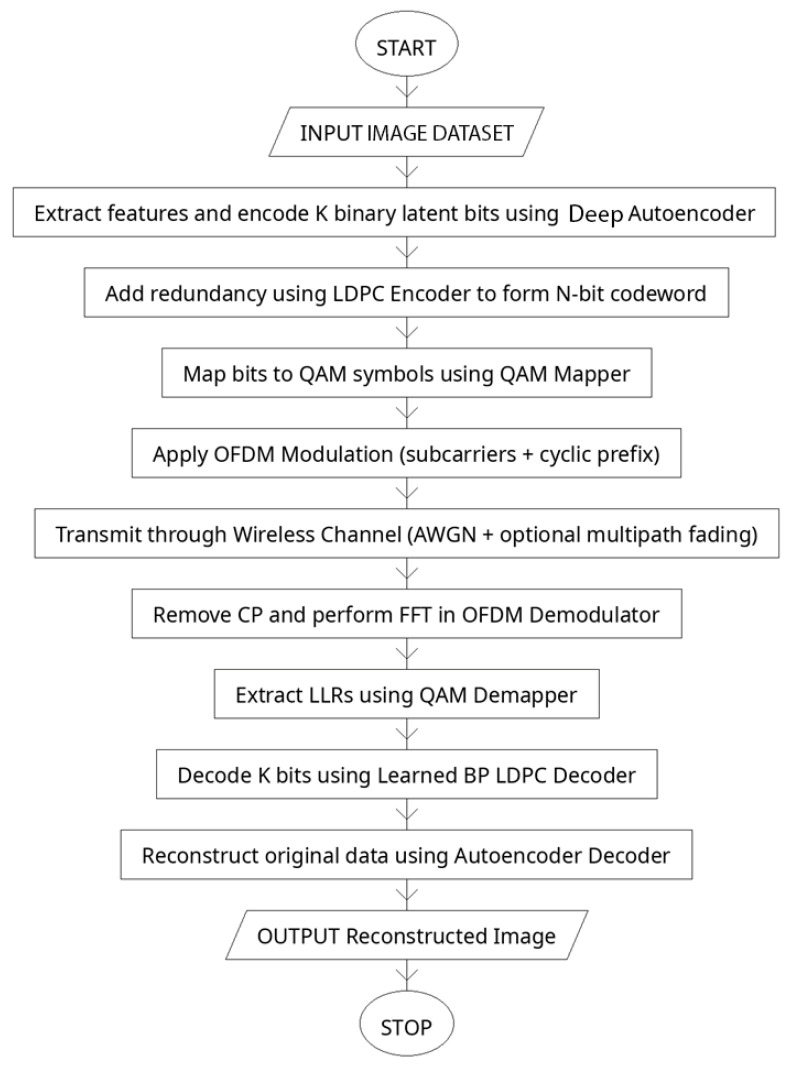
Architecture of the Proposed DAE–LDPC–OFDM Transceiver with Learned Belief Propagation Decoder.

**Figure 4 sensors-25-06776-f004:**
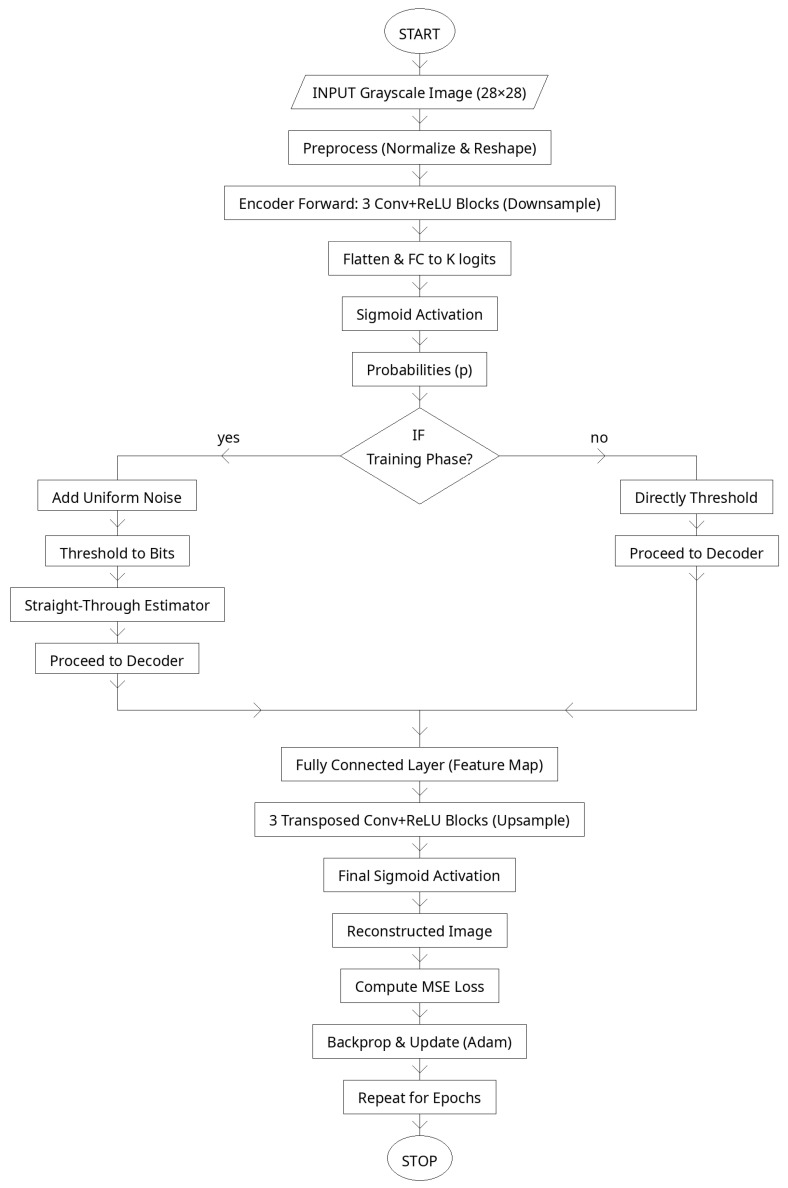
Deep Autoencoder Design (Binary Latent).

**Figure 5 sensors-25-06776-f005:**
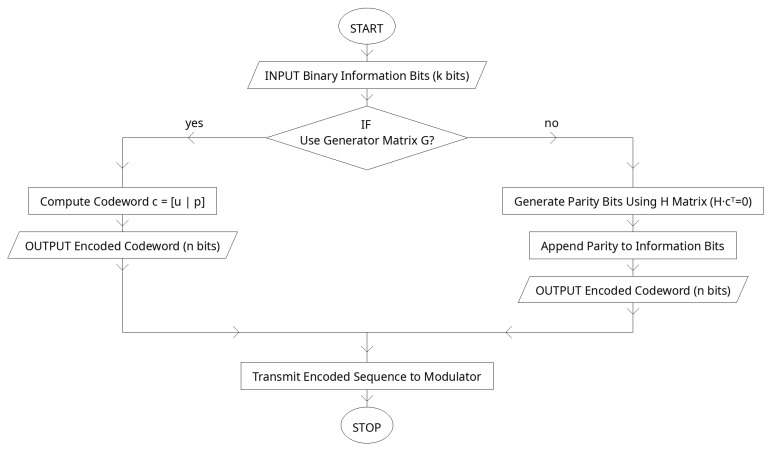
LDPC Encoding process.

**Figure 6 sensors-25-06776-f006:**
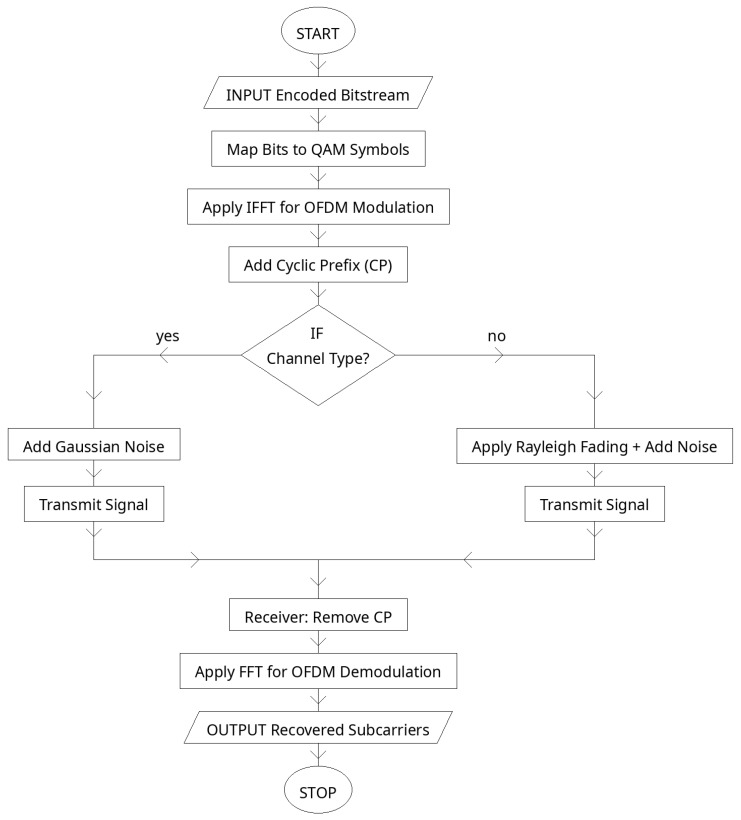
OFDM Modulation and Channel Model.

**Figure 7 sensors-25-06776-f007:**
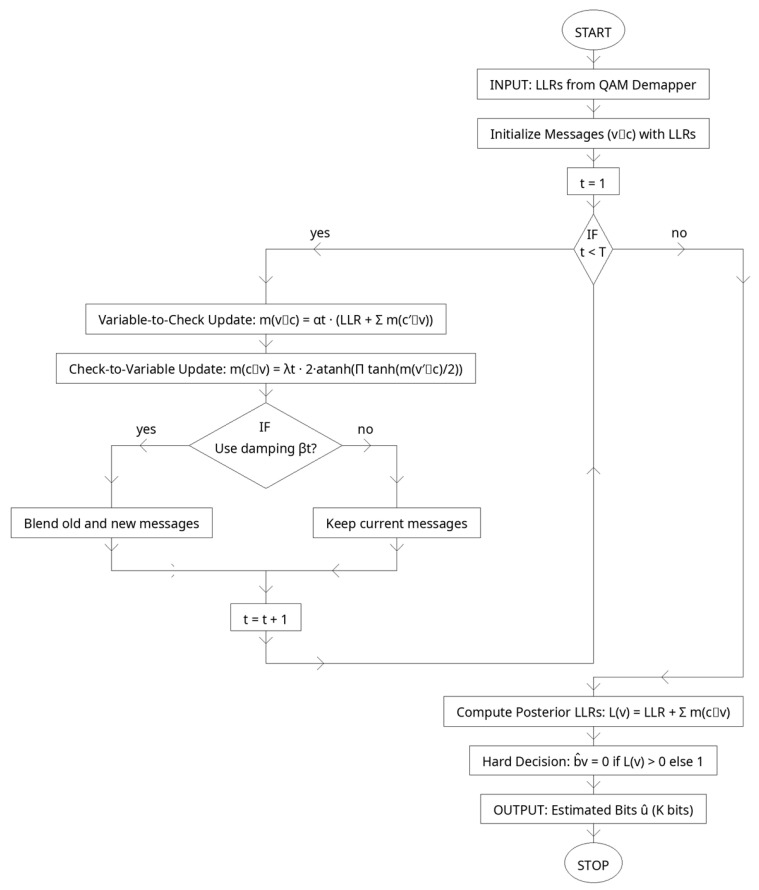
Learned Belief Propagation Decoder.

**Figure 8 sensors-25-06776-f008:**
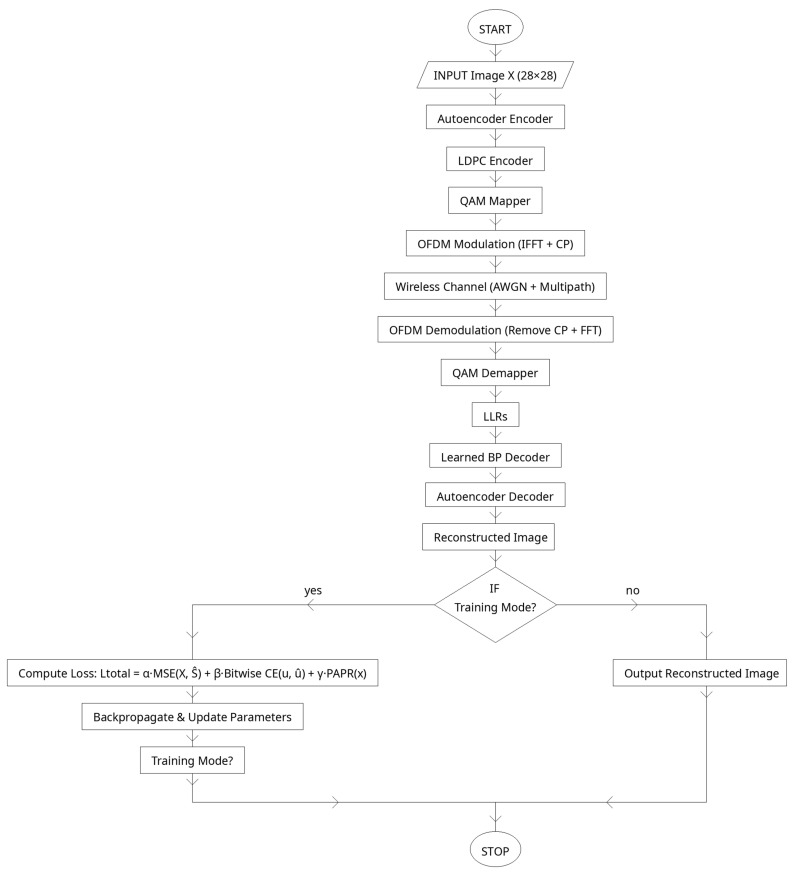
End-to-End Workflow and Loss Function.

**Figure 9 sensors-25-06776-f009:**
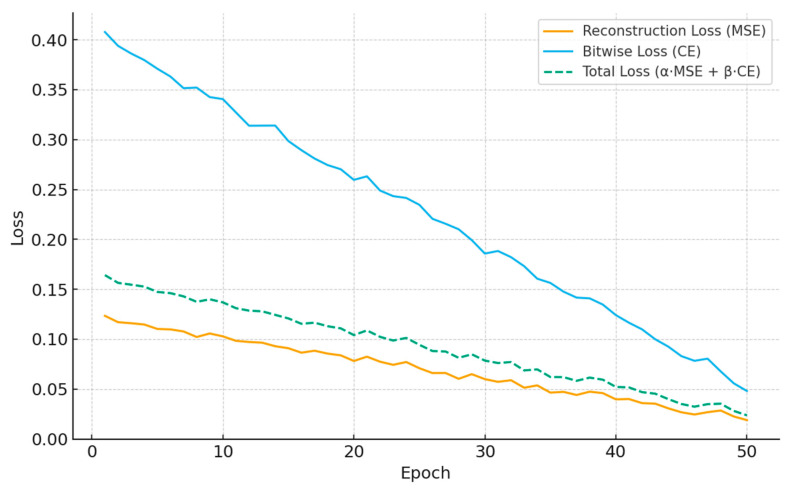
Training convergence of the proposed end-to-end system over 50 epochs.

**Figure 10 sensors-25-06776-f010:**
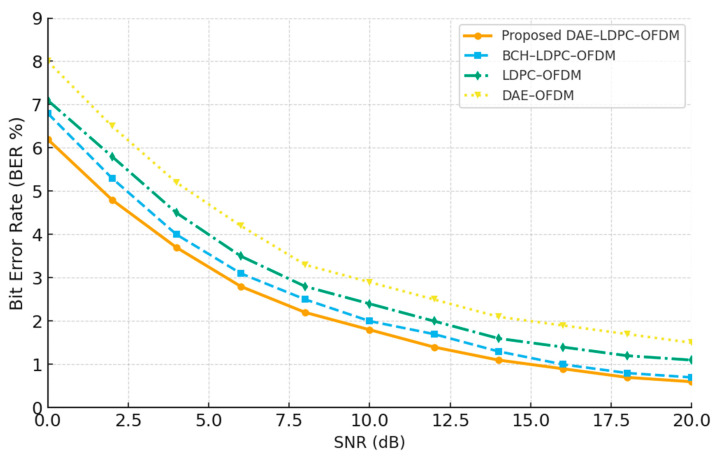
BER vs. SNR for DAE–LDPC–OFDM, DAE–OFDM, LDPC–OFDM, and BCH–LDPC–OFDM.

**Figure 11 sensors-25-06776-f011:**
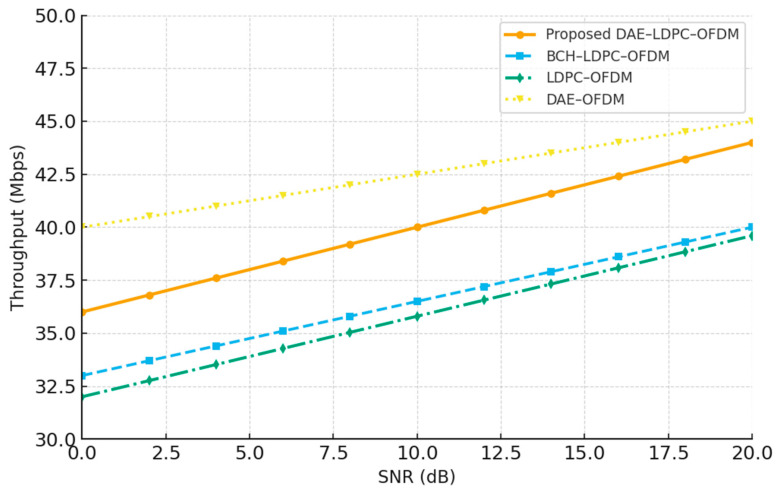
Throughput vs. SNR for DAE–LDPC–OFDM, DAE–OFDM, LDPC–OFDM, and BCH–LDPC–OFDM.

**Figure 12 sensors-25-06776-f012:**
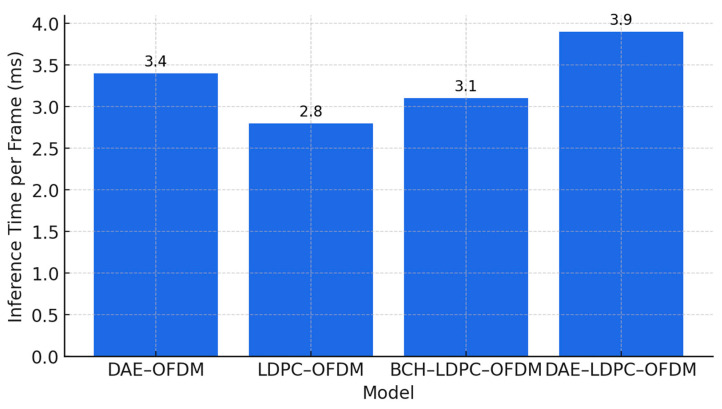
Runtime Distribution of Major Sub-Blocks in the DAE–LDPC–OFDM Transceiver.

**Figure 13 sensors-25-06776-f013:**
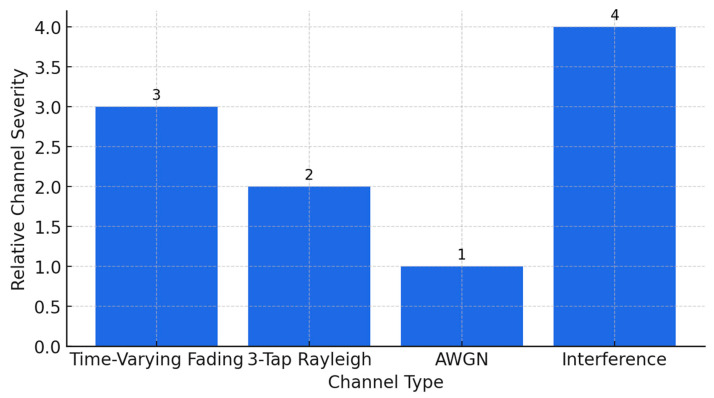
Extended Channel Scenarios Overview (AWGN, Rayleigh, Time-Varying, Interference).

**Figure 14 sensors-25-06776-f014:**
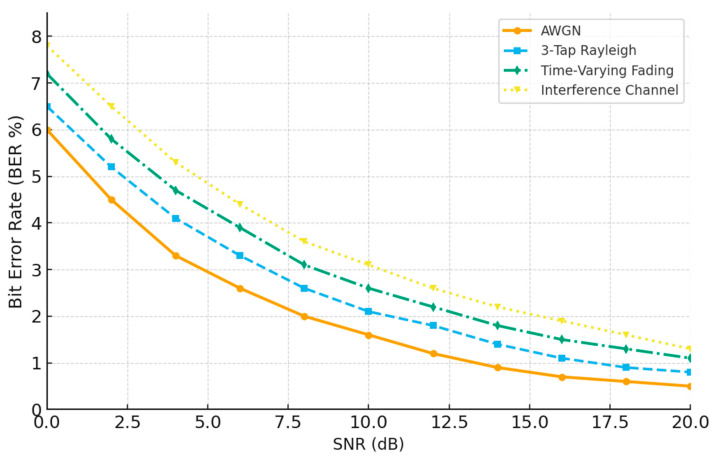
BER vs. SNR under Different Channel Models.

**Figure 15 sensors-25-06776-f015:**
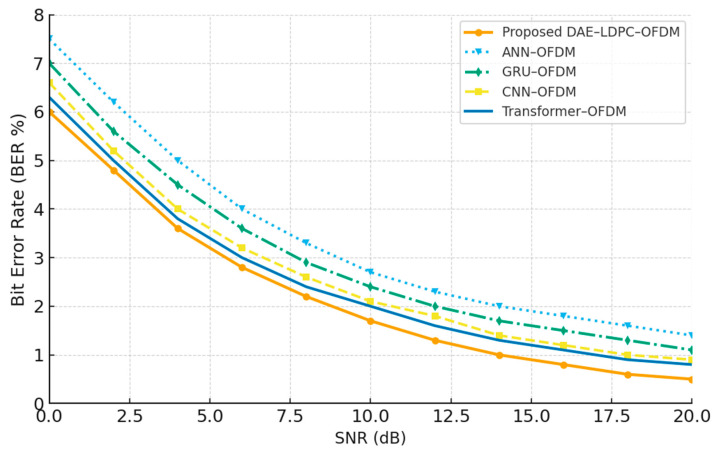
BER vs. SNR Comparison of DAE–LDPC–OFDM with ANN, GRU, CNN, and Transformer Models.

**Figure 16 sensors-25-06776-f016:**
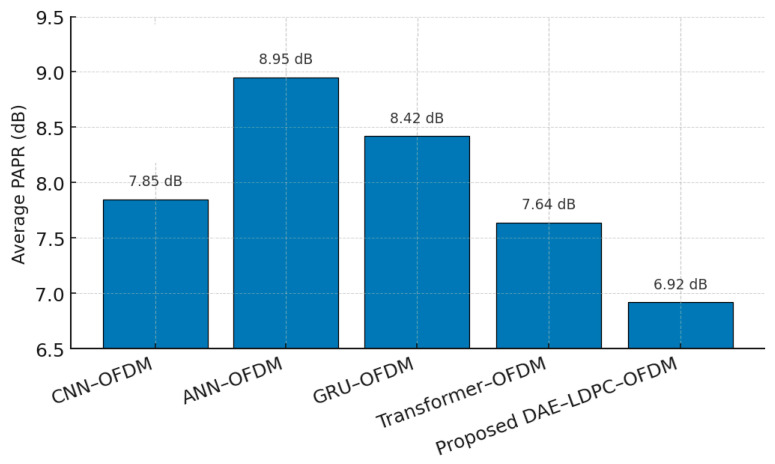
PAPR Reduction Performance Comparison across ANN, GRU, CNN, Transformer, and DAE–LDPC–OFDM Models.

**Figure 17 sensors-25-06776-f017:**
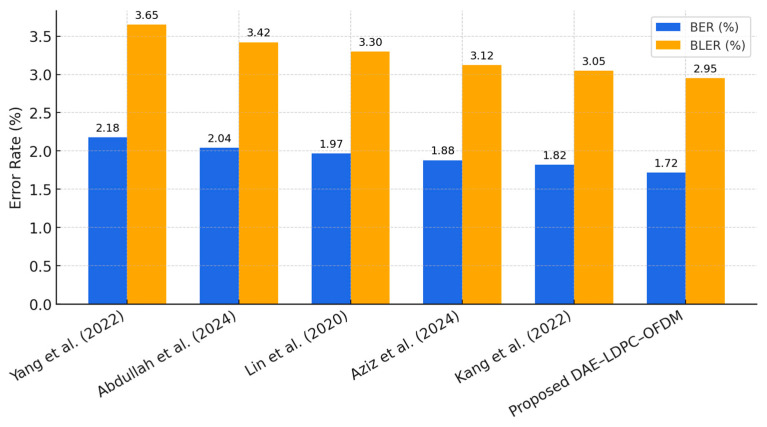
Comparison of BER and BLER Performance at 10 dB SNR for Recent Deep Learning–Based OFDM Systems [3,10,15,26,27].

**Table 1 sensors-25-06776-t001:** Summary of related works.

Ref	Focus/Contribution	Methodology	Key Findings	Limitations
[4]	Semantic feature-based communication for image transmission	Feature-driven semantic encoder–decoder over wireless channels	Reduced bandwidth use and maintained visual fidelity	Limited to semantic-level images; not generalized to complex signals
[5]	Layered HARQ with LDPC-coded modulation	Multi-layer error-control coding using LDPC and adaptive retransmission	Improved reliability and throughput	Increased system complexity
[6]	Integrated CRC and polar coding	Successive cancellation decoding combined with CRC	High detection and correction efficiency	Computational overhead for large data sizes
[7]	Neural lossy source coding	MLP-based compression framework	High compression with minimal distortion	Poor interpretability; requires tuning
[8]	Low-noise RF-PA supply modulator	Analog design for FDD systems	Improved linearity and energy efficiency	Hardware complexity
[9]	LDPC decoding in optical fronthaul	LDPC-coded WDM-PON over FSO channels	Enhanced reliability under turbulence	Sensitivity to atmospheric conditions
[10]	Asymmetrical AE for PAPR reduction	Deep learning-based encoder–decoder imbalance	Significant PAPR and BER reduction	Requires customization per application
[11]	Hierarchical Variational AE for image transmission	Probabilistic latent representation	Adaptive compression and robust reconstruction	High computational cost
[12]	LSTM-AE for VLC	Temporal feature extraction with recurrent AE	Effective PAPR suppression	Limited to VLC systems
[13]	Sparse AE for CSI feedback	Joint sparse learning for MIMO systems	Reduced feedback load with high accuracy	Complexity in sparse optimization
[14]	Stacked AE for PAPR reduction	Multi-layer AE mapping in VLC	Improved signal quality and lower PAPR	Moderate computational complexity
[15]	CNN-AE for maritime OFDM	End-to-end learning for channel estimation	High reliability in marine environments	Requires large-scale retraining
[16]	Comprehensive AE survey	Review of AE types and applications	Identified trends: sparse, variational, hybrid AE	Lacks experimental validation

**Table 2 sensors-25-06776-t002:** Specifications of Image Transmission Datasets.

Dataset	Image Size (Pixels)	Channels	Total Images	Type
CIFAR-10	32 × 32	RGB	60,000	Natural images
CIFAR-100	32 × 32	RGB	60,000	Natural images
Fashion-MNIST	28 × 28	Grayscale	70,000	Clothing items
CelebA	178 × 218	RGB	202,599	Human faces
Tiny-ImageNet	64 × 64	RGB	120,000	Object images
Kodak	768 × 512	RGB	24	Natural photographs
DIV2K	2040 × 1080 (avg.)	RGB	800	High-resolution natural images

**Table 3 sensors-25-06776-t003:** Deep Autoencoder Parameters.

Layer	Type	Input Size	Output Size	Kernel	Activation	Operation
1	Convolution + ReLU	1 × 28 × 28	16 × 14 × 14	3 × 3, stride 2	ReLU	Downsampling
2	Convolution + ReLU	16 × 14 × 14	32 × 7 × 7	3 × 3, stride 2	ReLU	Feature extraction
3	Convolution + ReLU	32 × 7 × 7	64 × 4 × 4	3 × 3, stride 2	ReLU	Deep feature encoding
4	Flatten + Fully Connected	64 × 4 × 4	K	—	Sigmoid	Latent projection
5	Fully Connected	K	64 × 4 × 4	—	ReLU	Latent expansion
6	Transposed Convolution + ReLU	64 × 4 × 4	32 × 7 × 7	3 × 3, stride 2	ReLU	Upsampling
7	Transposed Convolution + ReLU	32 × 7 × 7	16 × 14 × 14	3 × 3, stride 2	ReLU	Feature reconstruction
8	Transposed Convolution + Sigmoid	16 × 14 × 14	1 × 28 × 28	3 × 3, stride 2	Sigmoid	Final output reconstruction

**Table 4 sensors-25-06776-t004:** LDPC Parameters (Proposed System).

Parameter	Symbol	Value	Notes
Information length	K	128	Equals AE latent size
Codeword length	N	256	Matches OFDM mapping (see $3.4)
Parity length	M=N−K	128	Number of parity constraints
Code rate	R=K/N	0.5	Half-rate code
Var. node degree (avg.)	$d_{v}$	≈3	Sparse column weight of H
Check node degree (avg.)	$d_{c}$	≈6	Implied by $Nd_{v}-Md_{c}$
Parity-check matrix	H	128×256	Sparse; arranged as (d,mathbf{A});
Generator matrix	G	128×256	Systematic ([J. Jmathbf(II)]);
BP decoding iterations	T	8	Unrolled, trainable scalars per iter.
Decoding module	-	Learned BP	Differentiable, end-to-end trained

**Table 5 sensors-25-06776-t005:** OFDM Parameters.

Parameter	Symbol	Value	Notes
Number of subcarriers	$N_{sc}$	64	Matches codeword-to-symbol mapping (N−256,16-QAM →64 symbols)
Cyclic prefix length	$N_{cp}$	16	Provides guard interval to mitigate |S| from multipath
Modulation order (QAM)	M	16	4 bits per symbol, Gray-coded constellation
Bits per OFDM symbol	${log}_{2}(M)$	4	Determines mapping granularity
Symbols per OFDM block	S	64	Equals $N_{sc}$, one codeword spans one OFDM block
Sampling factor (IFFT/FFT)	-	$1/\sqrt{N_{sc}}$	Normalization to preserve average power
Channel taps	L	3	Example: [1, 0.3+0.3j, 0.2+0.2j] multipath profile
Channel model	-	AWGN + multipath	Evaluated under both AWGN-only and fading channels

**Table 6 sensors-25-06776-t006:** Learned Belief Propagation Decoder Parameters.

Parameter	Symbol	Value	Notes
Number of iterations	T	8	Unrolled message-passing steps
Variable-to-check scaling	$\alpha_{t}$	Trainable	One learnable scalar per iteration, refines aggregation strength
Check-to-variable scaling	$\lambda_{t}$	Trainable	One learnable scalar per iteration, adjusts parity constraint strength
Damping factor (optional)	$\beta_{t}$	Trainable	Smooths updates between iterations, prevents oscillations
Parameter sharing	-	Per-iteration	Shared across edges, different across iterations
Initialization	-	From LLRs	Decoder initialized with demapper-provided log-likelihood ratios
Output decision	${\hat{b}}_{v}$	Hard threshold	Bit =0 if $L_{v}^{\left(T\right)}>0$, else bit =1
Loss function	-	BCE on info bits	Combined with AE reconstruction and optional PAPR regularization

**Table 7 sensors-25-06776-t007:** End-to-End Workflow and Loss Components.

Stage	Operation	Input → Output	Trainable Parameters
Deep Autoencoder Encoder	CNN downsampling + binarization (STE)	$X\in R^{28\times 28}\to u\in \{0,\;1{\}}^{K}$	Conv + FC weights
LDPC Encoder	Linear coding with generator matrix	$u\to c\in \{0,\;1{\}}^{N}$	Fixed (defined by G)
QAM Mapper + OFDM Mod	Symbol mapping, IFFT, cyclic prefix	$c\to x_{\mathrm{cp}\;}$	None
Wireless Channel	AWGN + multipath convolution	$x_{\mathrm{cp}\;}\to y$	None
OFDM Demod + QAM Demap	CP removal, FFT, LLR extraction	$y\to l_{v}$	None
Learned BP Decoder	Iterative trainable message passing	$l_{v}\to \overset{ˆ}{u}$	$\alpha_{t},\lambda_{t},\beta_{t}$
Deep Autoencoder Decoder	FC + deconvolutions	$\overset{ˆ}{u}\to \overset{ˆ}{X}$	FC + ConvT weights
Loss Function	Composite objective	$X,\;\overset{ˆ}{X},\;u,\;\overset{ˆ}{u}$	α, β, γ

**Table 8 sensors-25-06776-t008:** Loss Convergence of the Deep Autoencoder over 50 Epochs (MNIST Dataset).

Epoch	Reconstruction Loss (MSE)	Bitwise Loss (CE)	Total Loss (α = 1.0, β = 0.1)
1	0.11	0.4	0.15
5	0.095	0.36	0.135
10	0.085	0.32	0.12
15	0.075	0.285	0.11
20	0.068	0.25	0.1
25	0.06	0.22	0.092
30	0.053	0.19	0.085
35	0.047	0.16	0.078
40	0.041	0.13	0.07
45	0.036	0.1	0.062
50	0.03	0.075	0.055

**Table 9 sensors-25-06776-t009:** Training Convergence and Final Loss Comparison Across Image Transmission Datasets.

Dataset	Final MSE	Final CE	Final Total Loss	Convergence Epoch
CIFAR-10	0.03	0.075	0.055	50
Fashion-MNIST	0.028	0.071	0.052	48
Tiny-ImageNet	0.037	0.089	0.059	55
CelebA	0.04	0.095	0.063	58
DIV2K	0.045	0.11	0.067	60

**Table 10 sensors-25-06776-t010:** BER and BLER Comparison at 10 dB Across Image Transmission Datasets.

Dataset	DAE–OFDM BER (%)	LDPC–OFDM BER (%)	BCH–LDPC–OFDM BER (%)	Proposed DAE–LDPC–OFDM BER (%)	Proposed DAE–LDPC–OFDM BLER (%)
CIFAR-10	2.38	2.1	1.96	1.85	3.1
Fashion-MNIST	2.25	2.04	1.9	1.72	2.95
CelebA	2.51	2.26	2.1	1.98	3.25
Tiny-ImageNet	2.6	2.4	2.2	2.05	3.37
DIV2K	2.75	2.58	2.34	2.15	3.52

**Table 11 sensors-25-06776-t011:** Effective Data Rate and Code Rate Comparison among DAE–LDPC–OFDM, LDPC–OFDM, and BCH–LDPC–OFDM.

System	Compression Ratio (CR)	Code Rate (R = k/n)	Effective Data Rate (Mbps)	Spectral Efficiency (η, bits/s/Hz)
DAE–OFDM	1	1	54	6.75
LDPC–OFDM	1	0.75	40.5	5.06
BCH–LDPC–OFDM	1	0.68	36.7	4.58
Proposed DAE–LDPC–OFDM	0.7	0.75	43.8	5.47

**Table 12 sensors-25-06776-t012:** Total Runtime and Computational Complexity Comparison.

Model	Inference Time per Frame (ms)	Training Time (s/epoch)	FLOPs (×10^8^)	Parameter Count (×10^6^)
DAE–OFDM	3.4	11	0.9	0.84
LDPC–OFDM	2.8	—	0.7	0.22
BCH–LDPC–OFDM	3.1	—	0.8	0.36
DAE–LDPC–OFDM	3.9	18	1.2	1.05

**Table 13 sensors-25-06776-t013:** Channel Model Parameters for Extended Simulation Scenarios.

Channel Type	Delay Spread (µs)	Doppler Frequency (Hz)	Interference Power (dB)	Fading Type	Description
AWGN	—	—	—	None	Baseline white Gaussian noise channel
3-Tap Rayleigh	[0, 2, 5]	0	—	Flat Fading	Static multipath channel
Time-Varying Rayleigh	[0, 2, 5]	100	—	Fast Fading	Mobility and Doppler-induced fading
Interference Channel	[0, 3, 8]	50	−10	Selective Fading	Multipath with co-channel interference

**Table 14 sensors-25-06776-t014:** Quantitative Comparison of BER, BLER, Complexity, and Model Parameters.

Model	BER @10 dB (%)	BLER @10 dB (%)	FLOPs (×10^8^)	Parameters (×10^6^)
ANN–OFDM	2.45	3.9	0.65	0.35
GRU–OFDM	2.18	3.55	0.95	0.52
CNN–OFDM	1.97	3.3	1.1	0.68
Transformer–OFDM	1.89	3.22	1.45	0.94
Proposed DAE–LDPC–OFDM	1.72	2.95	1.2	1.05

**Table 15 sensors-25-06776-t015:** Comparison of PAPR Reduction Performance among Neural-Based OFDM Models.

Model	Average PAPR (dB)	PAPR Reduction (%)	Complexity Level	Remarks
ANN–OFDM	8.95	7.3	Low	Slight improvement via feature scaling
GRU–OFDM	8.42	11	Medium	Temporal gating smooths amplitude peaks
CNN–OFDM	7.85	16.9	Medium	Convolutional filters regularize subcarriers
Transformer–OFDM	7.64	19.1	High	Self-attention reduces local peak clustering
Proposed DAE–LDPC–OFDM	6.92	26.6	Moderate	Latent compression + redundancy smoothing

**Table 16 sensors-25-06776-t016:** Performance Comparison of Recent Deep Learning–Based OFDM Systems.

Model/Reference	Approach Summary	SNR (dB)	BER (%)	BLER (%)
Yang et al. [3]	Joint source–channel coding with OFDM integration	10	2.18	3.65
Abdullah et al. [10]	Asymmetrical AE for PAPR reduction in CP-OFDM	10	2.04	3.42
Lin et al. [15]	CNN-based channel estimation for OFDM	10	1.97	3.3
Aziz et al. [26]	Deep AE for multi-carrier communication	10	1.88	3.12
Kang et al. [27]	Deep pilot design and MIMO channel estimation	10	1.82	3.05
Proposed DAE–LDPC–OFDM	Deep autoencoder with LDPC for adaptive coding	10	1.72	2.95

## Data Availability

The original contributions presented in this study are included in the article. Further inquiries can be directed to the corresponding author.

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
