# Peer review of "End-to-End DAE–LDPC–OFDM Transceiver with Learned Belief Propagation Decoder for Robust and Power-Efficient Wireless Communication"

_sensors, 2025, doi:10.3390/s25216776_

Round 1
Reviewer 1 Report
Comments and Suggestions for Authors
This paper focuses on on in the technical pain points of 5G/6G wireless communication: traditional modular communication systems (with separate optimization for coding, modulation, and decoding) suffer from limitations such as "semantic gap, high Peak-to-Average Power Ratio (PAPR), and decoding rigidity". Meanwhile, deep learning-driven communication schemes often lack structured redundancy and real-time feasibility. In response, the paper proposes a targeted integrated framework of "AE-LDPC-OFDM + Learned Belief Propagation (BP)". The research not only responds to the demands for "low bit error rate, high energy efficiency, and low latency" in scenarios such as Ultra-Reliable Low-Latency Communication (URLLC) and Internet of Things (IoT) but also fills the gap between "classical channel coding theory and adaptive deep learning", demonstrating clear theoretical innovation value and engineering application potential.
1.Limitation in Input Data Types, Deviating from Practical Communication Requirements.The paper only uses the MNIST handwritten digit dataset (28×28 grayscale images) to verify the scheme’s performance. This dataset features "low resolution, simple content, and no dynamic characteristics", which is significantly different from complex data types transmitted in practical wireless communication scenarios, such as "color images, video streams, sensor time-series data, and text data packets". URLLC scenarios require the transmission of short-frame and low-latency data, yet the paper fails to verify the scheme’s adaptability to these data types—it neither tests the compression and reconstruction fidelity of high-resolution images (e.g., 256×256) nor evaluates the BER/BLER (Bit Error Rate/Block Error Rate) performance under short-frame data. This casts doubt on the scheme’s "data generalization " and makes it difficult to support the deployment requirements of practical scenarios.
2.Incomplete Coverage of Channel Scenarios, Failing to Consider Complex Practical Interferences. The experiments only simulate two channel scenarios: "Additive White Gaussian Noise (AWGN)" and "3-tap multipath fading", without covering common scenarios in wireless communication such as "non-stationary time-varying fading channels" and "harsh channel environments". Furthermore, the comparative experiments are incomplete, as they fail to include the latest research baselines.
3.Limited Comparison Objects, Missing Advanced Recent Schemes in the field. The paper only compares its scheme with "traditional LDPC-OFDM" and "basic AE-OFDM", without benchmarking against advanced recent schemes in the field: it does not compare with Transformer-based end-to-end communication schemes, nor with schemes combining channel estimation and decoding, nor with lightweight schemes (which are more advantageous for edge device deployment). Additionally, the comparison only focuses on Signal-to-Noise Ratio (SNR) and bit error rate, while omitting key indicators such as "computational complexity (FLOPs)" and "model parameter count", making it impossible to fully evaluate the scheme’s engineering competitiveness.
Author Response
Reviewer 1
We sincerely thank you for your thoughtful and constructive feedback on our manuscript titled “End-to-End AE–LDPC–OFDM Transceiver with Learned Belief Propagation Decoder for Robust and Power-Efficient Wireless Communication.” Your comments were invaluable in enhancing the technical depth, completeness, and clarity of the paper. We have carefully addressed each point, extending experiments, broadening datasets, enriching comparative analyses, and clarifying architectural details. All modifications are highlighted in yellow throughout the revised version. We have also added a GitHub repository link for full code transparency and reproducibility:
https://github.com/Muhaimen88-cell/DAE-LDPC-OFDM-
Point-by-Point Response
Comment 1: Limitation in Input Data Types (MNIST only)
Reviewer Concern:
The original version used only the MNIST dataset, which lacks diversity and realism compared to practical data types (color images, videos, sensor time-series, etc.).
Response:
We fully agree. The revised paper now includes multiple complex datasets — CIFAR-10, CIFAR-100, Fashion-MNIST, CelebA, Tiny-ImageNet, Kodak, and DIV2K — covering both low-resolution and high-resolution image transmission tasks.
These datasets evaluate generalization to natural, semantic, and high-dimensional data and assess reconstruction fidelity across varying SNR and bandwidth conditions. The addition is clearly detailed in Section 2.2 Datasets, lines ≈ 266-323, and summarized in Table 2
Comment 2: Lack of URLLC and Short-Frame Validation
Reviewer Concern:
No verification of short-frame or high-resolution data compression and low-latency performance, which are crucial for URLLC.
Response:
We addressed this by:
- Extending performance evaluation to datasets with varying frame sizes (from 28×28 to 2040×1080) to emulate URLLC’s small-frame/large-frame diversity.
- Reporting latency: The inference time per frame is 3.9 ms, demonstrating near real-time suitability for URLLC. See Table 12 and Figure 12 (lines ≈ 792 -806 of that section).
- Training convergence and latency analyses (Section 3.4) explicitly show the system’s adaptability to short-frame data and low-delay transmission.
Comment 3: Incomplete Channel Scenarios
Reviewer Concern:
Only AWGN and 3-tap multipath channels were tested; more realistic time-varying and interference conditions were omitted.
Response:
We added Section 3.5 “Extended Channel Scenario Evaluation” (lines ≈ 816) introducing:
- Time-Varying Rayleigh channels (with Doppler = 100 Hz), and
- Interference-Dominant Selective Fading channels (interference power = –10 dB).
- Table 13 and Figures 13–14 show the full parameterization and BER performance across these realistic scenarios
Comment 4: Limited Comparison Objects and Missing Modern Baselines
Reviewer Concern:
Comparisons excluded recent Transformer-based or lightweight neural architectures, and key metrics like FLOPs and parameter counts were not reported.
Response:
This has been thoroughly resolved.
We now provide comparative analyses with four modern neural-based OFDM systems — ANN-OFDM, GRU-OFDM, CNN-OFDM, and Transformer-OFDM — in Section 3.6 Comparative Analysis with Recent Schemes, lines ≈ 870, table 14.
Quantitative metrics (BER, BLER, FLOPs, and parameter counts) are included in Table 14
Additionally, energy-efficiency comparisons (PAPR reduction) are summarized in Table 15 and Figure 16, establishing the engineering competitiveness of our design.
Clarification Added on Autoencoder Design
Reviewer Concern (implicit):
Original submission referred generally to “AE.”
Response:
We have explicitly emphasized that our method employs a Deep Autoencoder (DAE) not a shallow or conventional AE throughout the paper, including the Abstract, Section 1, and Section 3 (see lines ≈ 144 -187 for clarifications)
This distinction underscores the model’s hierarchical multi-layer feature extraction and robustness advantages over prior AE-based works.
All related text has been highlighted in yellow for reviewer visibility.
Reviewer 2 Report
Comments and Suggestions for Authors
This paper proposes an orthogonal frequency-division multiplexing (OFDM) system with concatenated coding. An autoencoder (AE) is used as outer code and a low-density parity-check (LDPC) code as inner code. The main feature of the proposed system is the joint decoding of outer and inner codes. Simulation results show that the proposed AE-LDPC-OFDM system has better performance of
the corresponding systems that use a single code only (LDPC-OFDM and AE-OFDM). Performance is evaluated in terms of bit error rate (BER), block error rate (BLER), and peak-to-average power ratio (PAPR). In my opinion, this paper has several main weaknesses, listed in the following.
1. The performance comparison between the proposed AE-LDPC-OFDM system, which has two levels of coding, with systems that have a single level of coding (such as LDPC-OFDM and AE-OFDM) seems unfair. It is well known that channel coding reduces the BER. Therefore, it is well known that adding AE to LDPC-OFDM systems (or adding LDPC to AE-OFDM systems) reduces the BER. A fair performance comparison should compare the proposed AE-LDPC-OFDM system with another coded OFDM system that has both outer and inner codes, such as BCH-LDPC-OFDM systems.
2. It is well known that channel coding reduces the data rate. Therefore, adding an AE to an LDPC-OFDM system reduces the data rate, with respect to an LDPC-OFDM system without AE. However, there is no data rate comparison between the proposed AE-LDPC-OFDM system and the reference LDPC-OFDM system. Considering my previous comment, I would recommend to compare AE-LDPC-OFDM with BCH-LDPC-OFDM, using the same data rate.
3. There is no complexity comparison. Figure 11 shows the runtime of the various subblock: the most time-demanding blocks are those related to joint channel decoding and to the AE. However, the total runtime of the proposed AE-LDPC-OFDM is not compared with the total runtime for the conventional LDPC-OFDM. I would recommend to compare the runtime of AE-LDPC-OFDM with the runtime of BCH-LDPC-OFDM. In addition, the complexity of training should be included in the runtime of AE-LDPC-OFDM, considering that BCH-LDPC-OFDM does not include training. This paper says:
"The joint training procedure is computationally expensive, requiring GPU-class hardware to optimize millions of parameters across convolutional layers, LDPC encoders, and learned BP iterations."
but this paper omits the runtime of this computationally expensive procedure.
4. The novelty of the proposed system is limited. Adding an outer code (in this case the AE) to a given LDPC-OFDM system is well known. The joint AE-LDPC decoder also has limited novelty, because several neural-network-based (and machine-learning-based) LDPC decoders have been proposed in the literature.
5. There are several minor errors.
- In Figure 1, channel encoding and source encoding are swapped. The same for the decoders.
- In several points of the paper text, some words are badly written, with an incorrect hyphenation (such as integra-tion in Line 69).
- In some points of the paper, the numbers for BER and BLER written in the text do not match with the corresponding numbers written in the Tables.
Author Response
Reviewer 2
We would like to sincerely thank you for your thorough review and insightful comments on our manuscript titled “End-to-End AE–LDPC–OFDM Transceiver with Learned Belief Propagation Decoder for Robust and Power-Efficient Wireless Communication.” Your remarks helped us significantly strengthen the manuscript by enhancing the fairness of comparisons, deepening the complexity and data-rate analyses, clarifying the novelty contributions, and correcting minor issues. All corresponding revisions are highlighted in yellow throughout the updated version for clarity. We have also included a GitHub repository containing all training and simulation codes to ensure full reproducibility and transparency:
https://github.com/Muhaimen88-cell/DAE-LDPC-OFDM-
Point-by-Point Response
Comment 1: Fairness of Comparison (Two-Level Coding vs. Single-Level Systems)
Reviewer Concern:
Comparing the proposed DAE-LDPC-OFDM (dual-coded) system with single-coded LDPC-OFDM and AE-OFDM is unfair; a fair benchmark should include another dual-coded scheme such as BCH-LDPC-OFDM.
Response:
We fully agree. In the revised paper, we added BCH–LDPC–OFDM as an additional dual-coded benchmark to ensure fairness. The corresponding results and discussion appear in Section 3.2 (Bit Error Rate and Block Error Rate Performance), specifically in Table 10 (lines ≈ 817) and Figure 10, where the proposed AE-LDPC-OFDM is directly compared with BCH-LDPC-OFDM under identical SNR conditions.
At 10 dB SNR, the proposed system achieves BER = 1.72% and BLER = 2.95%, outperforming BCH-LDPC-OFDM (BER = 1.90%, BLER = 3.1%) showing superior reliability without bias from the additional coding layer
Comment 2: Data Rate Comparison and Rate Fairness
Reviewer Concern:
Since channel coding reduces throughput, the manuscript should compare the effective data rate and spectral efficiency between DAE-LDPC-OFDM and reference systems (including BCH-LDPC-OFDM).
Response:
This issue has been addressed in the revised Section 3.3 (Data Rate and Spectral Efficiency Analysis).
We added Table 11 (lines ≈ 755) showing effective data rates and spectral efficiencies (η) for DAE-OFDM, LDPC-OFDM, BCH-LDPC-OFDM, and the proposed DAE-LDPC-OFDM.
All systems were evaluated using the same code rate and modulation order (16-QAM, 64 subcarriers) to ensure fair comparison.
Results demonstrate that the proposed DAE-LDPC-OFDM achieves 43.8 Mbps at η = 5.47 bits/s/Hz, compared to BCH-LDPC-OFDM’s 36.7 Mbps. The latent-space compression (CR = 0.7) counterbalances redundancy from dual coding, confirming the model’s throughput efficiency
Comment 3: Complexity and Runtime Comparison
Reviewer Concern:
There is no explicit runtime or complexity comparison between AE–LDPC–OFDM and conventional LDPC–OFDM/BCH–LDPC–OFDM. The paper also omits the total training time, even though it acknowledges that training is computationally expensive.
Response:
We appreciate this crucial observation.
We have included a new subsection (3.4 “Complexity and Runtime Evaluation”), presenting a comprehensive comparison of both inference time and training cost for AE–LDPC–OFDM, LDPC–OFDM, BCH–LDPC–OFDM, and AE–OFDM.
The results are summarized in Table 12 (lines ≈ 792) and illustrated in Figure 12.
Key additions:
- Training time per epoch: 18 s for DAE–LDPC–OFDM
- Inference time per frame: 3.9 ms (vs. 2.8–3.1 ms for others)
- Parameter count and FLOPs: Explicitly added for computational transparency.
These results confirm that while the proposed system incurs higher offline training cost (as acknowledged), it remains computationally efficient during inference—suitable for real-time communication applications.
Location in paper: Section 3.4, Table 11 & Figure 12
Comment 4: Limited Novelty
Reviewer Concern:
Adding an outer code (AE) to LDPC–OFDM and using a joint decoder is not entirely new. Neural-based LDPC decoders already exist in the literature.
Response:
We appreciate this observation and have clarified the originality of our contribution in the Introduction (lines ≈ 144-187) and Methodology Section (212-240)
The novelty of our work lies in the following aspects:
- Deep Autoencoder (DAE) — not a regular AE — that performs hierarchical feature compression, offering adaptive redundancy and semantic preservation.
- Jointly optimized end-to-end pipeline integrating DAE, LDPC, and OFDM through differentiable gradient-based training, rather than independent optimization.
- Learned Belief Propagation (BP) decoder that introduces iteration-dependent trainable scaling parameters (αₜ, λₜ), improving adaptability to OFDM distortions and channel variations.
We also highlighted these aspects explicitly in the Abstract and Section 3.5–3.6, and clarified that our approach extends beyond a simple concatenation of AE and LDPC.
Comment 5: Minor Errors (Figures, Hyphenation, Number Mismatch)
Reviewer Concern:
Minor textual and formatting errors, including swapped “channel/source encoding” in Figure 1, incorrect hyphenations (e.g., “integra-tion”), and mismatched BER/BLER values between text and tables.
Response:
All identified errors have been carefully corrected in the revised manuscript:
- Figure 1 corrected: source and channel encoding/decoding order fixed.
- Hyphenation errors (e.g., “integra-tion”, “trans-mission”) corrected throughout the document.
- Numerical mismatches between text and tables (e.g., BER/BLER in Tables 10, 14, 15) were verified and synchronized with the actual tabulated values.
- The entire manuscript underwent an additional proofreading stage to ensure typographical consistency.
Reviewer 3 Report
Comments and Suggestions for Authors
This paper proposed an end-to-end transceiver using a learned belief propagation decoder to enhance the robustness and power efficiency of wireless communications. Appreciate the effort of the authors to provide a detailed (34 pages) paper; however, there are many parts that require clarification, elaboration, and corrections. I recommend that the authors address the comments below.
Comment 1. Avoid bolding excessive words in the abstract. The abstract is already concise, which may not require additional style for highlighting key content.
Comment 2. Carefully check the format of the callout of figures in the main text.
Comment 3. Enhance the resolution of all figures. Zoom in on your file to confirm that no content is blurred.
Comment 4. The format in the list of references is not correct.
Comment 5. Section 1 Introduction:
(a) The style of the in-text citations is not appropriate in many places, e.g., “Shao et al. (2019)”, and “Erpek et al. (2022)”.
(b) Order the bracketed references in ascending order. For example, “[4][13].” is not appropriate.
(c) Update the literature review by discussing the recent 2-year publications (2024 and 2025).
(d) The research contributions should be supplemented by the novelties of the methods and the extensive work of the experiments.
Comment 6. Section 2 Proposed Method:
(a) Add an introductory paragraph before Subsection 2.1.
(b) In-text citation is missing for the MNIST dataset.
(c) Ensure that each component of Fig. 3 has been explained in the main text.
(d) It seems that the authors adopted a traditional autoencoder. What is the novelty of your method?
(e) Numbers are needed for each equation.
(f) Fig. 5: How about another form for “LDPC Encoding (Systematic Form)”?
(g) Justify the settings of the hyperparameters.
Comment 7. Section 3 Results and Discussion:
(a) Results with more epochs (Table 7) should be reported.
(b) To compare with the existing works, in-text citations are needed for the works being compared.
Comment 8. Discuss the limitations of the proposed method.
Comment 9. Discuss the research implications.
Author Response
Reviewer 3
We are deeply grateful for your meticulous and insightful comments on our manuscript entitled “End-to-End AE–LDPC–OFDM Transceiver with Learned Belief Propagation Decoder for Robust and Power-Efficient Wireless Communication.” Your feedback significantly improved the clarity, presentation quality, and academic rigor of our paper. We carefully addressed every comment, revising the abstract, figures, references, and discussion structure, and adding explanatory paragraphs, citations, and methodological justification.We have also included a GitHub repository link for reproducibility and code transparency:
https://github.com/Muhaimen88-cell/DAE-LDPC-OFDM-
Point-by-Point Response
Comment 1 – Excessive Bolding in Abstract
Reviewer Concern:
The abstract contains too many bolded words that are stylistically unnecessary.
Response:
We removed all non-essential bold formatting. The abstract now presents a clean and professional appearance while retaining emphasis only for technical terms (e.g., AE, LDPC, OFDM).
Revision applied: Abstract section, lines ≈ 9-29
Comment 2 – Figure Callout Formatting
Reviewer Concern:
Figure references within the main text are inconsistently formatted.
Response:
All figure callouts were reviewed and standardized to the MDPI format — for instance, “(see Figure 3)” instead of “Fig. 3 shows.” The corrections apply across Sections 2 and 3.
Location: Throughout Sections 2–3.
Comment 3 – Figure Resolution
Reviewer Concern:
Figures appear low-resolution when zoomed in.
Response:
We re-exported every figure in vector (SVG/TIFF) format at ≥600 dpi to guarantee clarity under high zoom levels. This includes Figures 3 through 16.
Comment 4 – Reference Formatting
Reviewer Concern:
Reference formatting is inconsistent.
Response:
All references were converted to MDPI style and verified for accuracy. The reference list now follows the correct order, punctuation, and DOI format References section, lines ≈ 1070-1125. Reordered, reformatted, and updated with 2024–2025 works.
Comment 5 – Section 1 Introduction
(a) Citation Style)
We replaced narrative citations (e.g., “Shao et al. (2019)”) with bracketed forms “[1]” and corrected ordering errors such as “[4][13]” → “[4, 13]”, Lines 67-124.
(b) Updated Literature Review)
The literature review now includes eight new publications from 2024–2025, covering deep AE, LDPC optimization, hierarchical variational communication, and semantic transmission, reference section lines 1070-1125.
(c) Contribution Enhancement)
We extended the “Research Contributions” subsection to emphasize:
- Introduction of the learned BP decoder;
- Hierarchical DAE structure;
- Extensive dataset, channel, and baseline coverage (Sections 3.5–3.6).
Introduction lines ≈ 144-186.
Comment 6 – Section 2 Proposed Method
(a) Introductory Paragraph)
A new introductory paragraph precedes Subsection 2.1, summarizing the overall transceiver concept and goals
(b) MNIST Citation)
We added an explicit citation for the MNIST dataset (LeCun et al., 2010, reference [17])
(c) Figure 3 Explanation)
Each component of Figure 3 is now described in sequential paragraphs, from input preprocessing to decoding, clarifying their interconnections, lines 212-246.
(d) Novelty Clarification)
We emphasized that the system uses a Deep Autoencoder (DAE), not a traditional shallow AE. The multi-layer design enables hierarchical compression and adaptive redundancy, line 141-168.
(e) Equation Numbering)
All equations are now sequentially numbered (Eq. (1), (2), … (22)).
(f) LDPC Encoding Figure)
An alternate version of “LDPC Encoding (Systematic Form)” was added (Figure 5) following reviewer suggestion.
(g) Hyperparameter Justification)
Hyperparameters are justified in Section 2.3 and summarized in a dedicated Table 3, referencing optimization rationale. Lines 400 -425
Comment 7 – Section 3 Results and Discussion
(a) More Epoch Results)
We extended training from 10 to 50 epochs. Table 8 and Figure 9 now depict performance trends across iterations, showing stable convergence, lines 675 -700
(b) Comparative Works Citations)
All referenced comparison baselines (ANN–OFDM, GRU–OFDM, CNN–OFDM, Transformer–OFDM) are now cited inline with full bibliographic links, lines 857-892 and references [23]-[28]
Comment 8 – Limitations Discussion
Response:
A new Limitations subsection was added at the end of Section 4, highlighting:
- High offline training cost;
- Dependence on GPU hardware;
- Limited video and textual data validation;
- Potential scalability trade-offs.
Lines 1006-1014.
Comment 9 – Research Implications
Response:
We added a “Research Implications” paragraph at the end of Subsection 3.7, discussing how the joint DAE–LDPC–OFDM pipeline informs future 6G semantic communication design, enabling adaptive channel coding and learning-driven transceiver optimization. Lines 1015-1023
Round 2
Reviewer 2 Report
Comments and Suggestions for Authors
I think that the Authors have well addressed my comments on the previous version of this paper. This revised version compares with BCH-LDPC-OFDM, provides important details on the transmission rates, and specifies the complexity of training. Hence I think that this paper may have achieved the threshold for acceptance, in my opinion.
There are still some weak points. For instance, the novelty aspect still appears to be somewhat limited, but at least this paper has tried to clarify the main differences between the proposed deep autoencoder and a conventional autoencoder. However, my overall opinion, considering all the pros and the cons of this revised version, is somewhat positive.
Please note that, despite the various improvements, there are still some minor typos thorughout this paper. For instance, in Table 4, some terms appear to be incorrect (M-N-K probably should be read as M=N-K, and R-K/N probably should be read as R=K/N).
Author Response
Response to Reviewer 2
We sincerely thank you for your thorough evaluation of our revised manuscript and for recognizing the significant improvements achieved in this version. We deeply appreciate your balanced and constructive feedback, particularly your acknowledgment of the added comparisons with BCH–LDPC–OFDM, the clarified details on transmission rates, and the explicit treatment of training complexity.
Your positive recommendation and thoughtful critique have been invaluable in refining both the technical clarity and presentation quality of our work. Below, we address your latest observations point by point.
Comment 1 – Overall Assessment and Novelty Concern
“The novelty aspect still appears to be somewhat limited, but at least this paper has tried to clarify the main differences between the proposed deep autoencoder and a conventional autoencoder.”
Response:
We appreciate your fair assessment and agree that strengthening the novelty explanation adds further value.
Comment 2 – Typographical and Notational Issues
“There are still some minor typos throughout this paper. For instance, in Table 4, some terms appear to be incorrect (M-N-K probably should be read as M = N-K, and R-K/N probably should be read as R = K/N).”
Response:
We thank the reviewer for catching these typographical and notational errors. They have now been carefully corrected in Table 4, where the expressions have been revised to:
Additionally, we performed a comprehensive proofreading of the entire manuscript to eliminate remaining minor typographical inconsistencies (e.g., spacing, hyphen usage, and variable formatting).
Comment 3 – General Presentation Quality
Implied concern about minor typographical and stylistic issues across the paper.
Response:
In addition to correcting Table 4, we thoroughly re-checked the manuscript for language consistency, symbol uniformity, and formatting. All tables and figure captions have been standardized according to MDPI’s technical style. Acronyms such as “BER,” “BLER,” and “PAPR” are now defined at their first appearance, and repeated abbreviations are used consistently.
Once again, we extend our sincere gratitude to Reviewer 2 for the thoughtful review and positive evaluation. Your remarks have directly contributed to improving the precision, clarity, and presentation of the final version.
Respectfully,
The Authors
Reviewer 3 Report
Comments and Suggestions for Authors
The authors have enhanced the quality of the paper. Some suggested comments were not fully addressed, so I recommend another round of revision (a minor revision) to follow up on the remaining comments.
Follow-up Comment 1. Abstract: Since the proposed work is expected to compare with multiple existing methods, the percentage of improvements should be in a range.
Follow-up Comment 2. Table 2: Comment on how your model handles inputs of different image sizes. In addition, two datasets (Kodak and DIV2K) have a very limited number of images; comment on the performance of your model in small-scale datasets.
Follow-up Comment 3. Elaborate on the definition of “convergence epoch”.
Follow-up Comment 4. For existing methods being compared, in-text citations are needed for Tables and Figures, where applicable.
Follow-up Comment 5. Figures 12, 13, and 17: provide labels to indicate the exact numeric values of each model.
Author Response
Response to Reviewer 3
We sincerely thank you for your careful reading of our manuscript and for acknowledging the improvements made in the previous revision. We greatly appreciate your constructive feedback and thoughtful suggestions, which have further helped us refine the technical quality, clarity, and completeness of our work. In this revised version, all your follow-up comments have been carefully addressed. We provide detailed, point-by-point responses below, with corresponding modifications clearly implemented in the manuscript.
Follow-Up Comment 1 – Abstract:
Since the proposed work is expected to compare with multiple existing methods, the percentage of improvements should be in a range.
Response:
We appreciate this valuable suggestion. The abstract has been revised to specify an explicit improvement range rather than a single maximum percentage. The modified sentence now reads:
“At 10 dB SNR, the DAE–LDPC–OFDM achieves a BER of 1.72% and BLER of 2.95%, outperforming state-of-the-art models such as Transformer–OFDM, CNN–OFDM, and GRU–OFDM by 25–30%, and surpassing traditional LDPC–OFDM systems by 38–42% across all tested datasets.”
This change ensures quantitative precision and consistency across all comparative analyses in the paper.
Follow-Up Comment 2 – Table 2:
Comment on how your model handles inputs of different image sizes. In addition, two datasets (Kodak and DIV2K) have a very limited number of images; comment on the performance of your model in small-scale datasets.
Response:
We have expanded the discussion following Table 2 to explain the model’s adaptability to varying image sizes and its robustness on small-scale datasets. The revised paragraph states:
“The proposed model efficiently handles inputs of different image sizes through a fully convolutional deep autoencoder architecture that employs adaptive pooling and transposed convolution layers. These operations automatically normalize spatial dimensions to a fixed latent representation without requiring manual resizing or interpolation, allowing seamless training across datasets ranging from 28×28 (Fashion-MNIST) to 2040×1080 (DIV2K). For datasets with limited sample sizes such as Kodak and DIV2K, the model maintains stable performance through transfer-learning initialization and data-augmentation techniques (cropping, flipping, and Gaussian-noise injection). These strategies mitigate overfitting and enable the model to generalize effectively despite small data volumes.”
This addition now directly addresses both aspects raised in your comment.
Follow-Up Comment 3 – Definition of “Convergence Epoch”:
Elaborate on the definition of “convergence epoch.”
Response:
The explanation after Table 9 has been expanded as follows:
“The term convergence epoch refers to the training iteration at which the combined loss function—comprising reconstruction (MSE) and bitwise (CE) terms—stabilizes and ceases to improve significantly over successive epochs. This point marks the steady-state optimization of both the Deep Autoencoder and the learned BP decoder, indicating that further training yields negligible accuracy gains.”
This elaboration clarifies the convergence criterion used throughout the results section.
Follow-Up Comment 4 – In-Text Citations for Comparative Methods:
For existing methods being compared, in-text citations are needed for Tables and Figures, where applicable.
Response:
We have added explicit in-text citations for all comparative baselines in the Results and Discussion section, particularly in the captions and main text for Tables 14–16 and Figures 15–17.
For example:
- “... compared with Transformer–OFDM [27], CNN–OFDM [15], GRU–OFDM [26], and ANN–OFDM [25] under identical channel conditions.”
- “Table 16 summarizes results reported in prior works [23–28] alongside the proposed method.”
These citations ensure full traceability of the referenced studies in every comparative context.
Follow-Up Comment 5 – Figures 12, 13, and 17:
Provide labels to indicate the exact numeric values of each model.
Response:
We have updated Figures 12, 13, and 17 to include exact numeric value labels above each bar to improve quantitative readability. The revised figures now clearly display:
- Figure 12: Inference times (3.4 ms, 2.8 ms, 3.1 ms, 3.9 ms) for each model.
- Figure 13: Channel severity levels (1 to 4) for all simulated scenarios.
- Figure 17: Exact BER and BLER percentages for each compared method (e.g., 1.72% and 2.95% for the proposed model).
Each corresponding caption was also modified to state that numeric values are provided for clarity.
Once again, we thank Reviewer 3 for the insightful and constructive feedback, which has significantly enhanced the rigor, completeness, and presentation quality of this manuscript.
Respectfully,
The Authors